# Cdhr1a and pcdh15b may link photoreceptor outer segments with calyceal processes revealing a potential mechanism for cone-rod dystrophy

Meet K Patel, Warlen Pereira Piedade, Jakub K Famulski*

Department of Biology, University of Kentucky, Lexington, United States

## eLife Assessment

This **valuable** study investigates the interaction of two integral membrane proteins (Cdhr1a and Pcdh15b) and their roles in cone-rod dystrophy. **Convincing** evidence using loss-of-function mutants demonstrates clearly that both proteins are required for cone maintenance and survival. Although some evidence (Western blots and cell aggregation assays) demonstrates Cdhr1a and Pcdh15b can physically interact, there is insufficient evidence to support the subcellular localization and the proposed heterodimeric interaction of the two proteins from distinct subcellular compartments in cone photoreceptors.

*For correspondence:
jakub.famulski@uky.edu

**Abstract** Cone-rod dystrophy (CRD) is a macular degeneration disorder characterized by initial cone cell degeneration. Mutations in CDHR1, a photoreceptor-specific cadherin, have been found to be associated with the incidence of CRD. While studying the function of CDHR1, we observed that the localization of the zebrafish homologue, cdhr1a, resembles that of calyceal process (CPs). When co-labeling CPs using pcdh15b, we observed that cdhr1a, in the outer segment (OS), juxtaposes with pcdh15b, found in the CP. Similar localization patterns were detected in human, macaque, xenopus, ducks, gerbil, and mouse. Using immunoprecipitation and K652 cell aggregation assays, we demonstrate that pcdh15b and cdhr1a can interact and thus potentially link the OS and CP. To analyze the consequences of OS-CP interactions in CRD, we established a *cdhr1a* mutant line (*cdhr1a*$^{fs*146}$). Homozygous *cdhr1a*$^{fs*146}$ mutants exhibit minor cone OS defects starting at 15 dpf and severe OS disruption and cell loss by 3 months. Shortening of CPs coincided with cone OS defects which were significantly exacerbated when combined with the loss of pcdh15b. Rod OS defects were mild and delayed until 3–6 months. In conclusion, we propose that cdhr1a and pcdh15b function to link cone OSs with CPs and maintain OS integrity.

## Introduction

Inherited retinal diseases (IRDs) are a group of disorders that can severely affect vision and even lead to blindness affecting over 450,000 individuals in the US alone (*Ackland et al., 2017*). Cone-rod dystrophy (CRD) is a subset of IRD that leads to cone photoreceptor cell (PRC) degeneration followed by decline of rod PRCs (*Ackland et al., 2017*; *Bowne and Diager, 2020*). The disorder generally stems from mutations in one of the over 30 gene candidates. Function of a putative CRD candidate, CDHR1 has been known to clinically correlate with recessive CRD (*Arno et al., 2016*; *Duncan et al., 2012*; *Stingl et al., 2017*). CDHR1, a photoreceptor-specific cadherin, was first investigated by Rattner and colleagues in 2001 (previously PCDH21) where it was shown that a mouse KO resulted in early onset

retinal degeneration, at 1 month and severe degeneration by six (*Rattner et al., 2001*). Subsequent studies have shown that initial retinal degeneration involves loss of cones and subsequently rods (*Yusuf et al., 2021*). Molecularly, in the mouse model, it's been shown that CDHR1 localizes at the leading edge of OS discs and forms cadherin-based junctions to an unknown partner in the inner segment (*Rattner et al., 2001*; *Burgoyne et al., 2015*). These connections have been hypothesized to regulate nascent rod OS disc release into the mature OS (*Rattner et al., 2004*). However, CDHR1 function in cones has not been postulated. In *Xenopus*, absence of Cdhr1 results in overgrown OS discs at the basal OS, thereby disrupting the normal OS (*Carr et al., 2021*). In humans, CDHR1 mutations, both truncations and missense mutations, have been shown to correlate with late onset CRD (*Stingl et al., 2017*; *Ba-Abbad et al., 2021*; *Bolz et al., 2005*; *Ostergaard et al., 2010*). Clinically, these studies identify various pathogenic variants and variants of unknown significance; however, the exact mechanism behind CDHR1-based CRD pathogenesis remains unknown.

CDHR1, like other cadherins, encodes an intracellular domain, a transmembrane domain, and six evolutionarily conserved extracellular cadherin domain repeats (EC repeats; *Rattner et al., 2001*; *Rattner et al., 2004*). The cytoplasmic domains of classical cadherins have been shown to bind beta-catenin, which anchors the cadherin to various cytoskeletal elements such as actin (*Aberle et al., 1996*). Interestingly, the cytoplasmic domain of Cdhr1 diverges considerably from other cadherins and is predicted to not possess any beta catenin or cytoskeletal binding activity (*Rattner et al., 2004*; *Yang et al., 2008*). ECs of most cadherins possess negatively charged sequence motifs, which are involved in $Ca^{2+}$ binding (*Hirano et al., 1987*). These ECs form either homophilic or heterophilic junctions in trans conformation and thereby mediate inter/intracellular connections (*Takeichi et al., 1986*). Protocadherins generally consist of six to seven EC repeats and are predominantly observed with high diversity in various neuronal populations (*Pancho et al., 2020*). In neurons, protocadherins are responsible for synaptogenesis, neuronal specificity, and formation of mechanical junctions (*Halbleib and Nelson, 2006*). Based on mouse studies, CDHR1 forms connections with the IS to what appears to be the periciliary ridge and an extension of the IS appearing to be the calyceal process (*Burgoyne et al., 2015*).

Calyceal processes (CPs) were first described by *Cohen, 1961* (as calyx of a flower) surrounding the basal OS discs in rhesus macaque and pigeon photoreceptors (*Cohen, 1961*). Pioneering work in the 1970–80 s showed these processes are actin-based microvilli projections of the IS found in both rods and cones of various vertebrates including zebrafish and humans (*Burnside, 1978*; *Chaitin and Bok, 1986*). In macaque PRCs, thicker, longer, and more numerous CPs were observed surrounding cones versus rods (*Sahly et al., 2012*). Based on their proximity to the OS, CPs were hypothesized to structurally support the OS discs in both rods and cones (*Sharkova et al., 2024*). However, functional evidence of their potential aid to the OS remains scarce. Previous studies also show how CPs are absent and a vestigial remanence of periciliary ridge extension is observed in popular rodent models such as mice and rats, which may suggest a lack of functional evidence of CPs (*Sahly et al., 2012*; *Cohen, 1960*). Studies in macaque PRCs indicated localization of various Usher syndrome 1 (Ush1) proteins to the calyceal processes (*Sahly et al., 2012*). Clinically, Usher syndrome 1 is categorized by loss of hearing and vision due to mutations in one of the six Ush1 genes (*Mathur and Yang, 2015*). Of these six Ush1 genes, Pcdh15 has been shown to make heterophilic cadherin-based junctions in stereocilia of hair cells with another Ush1 gene, Cadherin 23 (Cdh23; *Kazmierczak et al., 2007*). Thereby, they form a mechanical bridge between adjacent stereocilia. Mechanical deflection of stereocilia due to sound waves changes the tension of Pcdh15-Cdh23 based interactions, which then triggers the mechano-electrical transduction channel for signal propagation (*Goldberg, 2006*; *McPherson, 2018*). Loss of functional Pcdh15 results in Usher syndrome type 1 f, entailing the loss of hearing and vision (*Lelli et al., 2010*). In PRCs, Pcdh15 has been unequivocally shown to be localized in the CPs (*Sahly et al., 2012*; *Miles et al., 2021*; *Schietroma et al., 2017*). Interestingly, loss of pcdh15b function in zebrafish results in an abnormal OS phenotype and PRC degeneration (*Miles et al., 2021*). However, the mechanism of pcdh15b-based PRC pathogenesis remains unknown. Noting the expression of two retinal cadherins in juxtaposing structures, CDHR1 in the OS and PCDH15 in the CP, we sought out to investigate whether CDHR1 and PCDH15 interact to link the OS discs and the CP.

In our current study, we demonstrate that in zebrafish PRCs, cdhr1a and pcdh15b juxtapose along the OS and CP, suggesting they form functional connections. Importantly, we show that this localization pattern is evolutionarily conserved up to and including humans. Furthermore, we confirm

pcdh15b-cdhr1a interactions via immunoprecipitation and K562 cell aggregation. While modeling CRD using a cdhr1a loss-of-function mutant line, we report progressive defects in CP morphology correlating specifically with cone degeneration. Taken together, we propose that loss of cdhr1a function disrupts OS-CP junctions and subsequently leads to CRD.

## Results

### Cdhr1a localization in the OS suggests an interaction with the calyceal process

As previously mentioned, CDHR1 has been shown to localize to the inner/outer segment boundary in several species, and in mouse rod cells, it was shown to extend from newly formed discs and form linkages with the inner segment. The connecting partner had yet to be determined. To expand our understanding of CDHR1 localization, we performed IHC for cdhr1a and imaged the sections using high-resolution confocal microscopy. When focusing on cone cells, we observed that cdhr1a was not restricted to the IS/OS boundary; instead, cdhr1a localization was found along the edges of OSs (*Figure 1A*). Conversely, in rod cells, we observed that cdhr1a localization was restricted to a much smaller region of the rod OS and closely associated with the IS, more reminiscent of the mouse cryo-EM studies (*Figure 1B*). The cone pattern of localization was reminiscent of recent work showcasing the actin-based IS extensions called calyceal processes (CP). While little is known about the exact molecular function of CPs, recent studies have shown that the Usher-syndrome-associated protein pcdh15b, also a protocadherin, localizes to the CP in zebrafish and in non-human primates. After examining zebrafish retinal expression of 5 key Usher proteins (Pcdh15b, Cdh23, Myo7a, Ush1c, and Ush1g), we only detected pcdh15b in the PRC layer (*Figure 1—figure supplement 1*). As expected, pcdh15b localization outlined the presumptive CPs in both cone and rod cells (*Figure 1A' and B'*). Thus, we next directly compared cdhr1a localization to that of pcdh15b and excitedly observed a close association between the two signals in both rods and cones (*Figure 1A'' and B''*). Interestingly, the localization of cdhr1a and pcdh15b did not appear to directly overlap, thus suggesting that the two proteins may juxtapose one another. To examine their localization at higher resolution, we turned to the super resolution technique structured illumination microscopy (SIM). Using SIM, we confidently confirmed that cdhr1a and pcdh15b signals juxtapose along the OS and the CP, respectively (*Figure 1C*). Additionally, we also imaged the localization of actin, which is expected to be restricted to the CP. When comparing cdhr1a and actin signals, we observed a juxtaposition rather than complete co-localization (*Figure 1D*). On the other hand, when comparing pcdh15b and actin, we observed near total co-localization, suggesting both are localized in the CP (*Figure 1E*). These localization patterns were visualized in both lateral and transverse views (*Figure 1F–H*). Taken together, this data suggested that cdhr1a in the OS and pcdh15b in the CP may be interacting. To further validate cdhr1a localization and the potential for interaction with pcdh15b, we employed immunogold-TEM. Using gold-labeled secondary antibodies, we were able to co-localize cdhr1a and pcdh15b at the interphase between outer segments and CPs, thus validating our immunohistochemistry results (*Figure 1I*). Based on these observations, and the fact that both cdhr1a and pcdh15b are from the cadherin family with heterophilic interaction tendencies, we hypothesize that cdhr1a connects the OS to the CP by cadherin-based interactions with pcdh15b (*Figure 1J*). Furthermore, we predict that cdhr1a-pcdh15b interactions span the majority of the outer segment in cones but only connect newly forming discs in rod outer segment. To determine if this hypothesis pertains only to teleost fish, we also examined localization of CDHR1 and PCDH15 across various vertebrate species including frog, duck, mouse, rat, gerbil, spiny mouse, macaque, and human (*Figure 2*). In all the species examined, we observed a close localization or juxtaposition of Cdhr1 and Pcdh15 signals indicating that their association is likely to be evolutionarily conserved. Furthermore, as was observed in zebrafish, in all the species examined, the vertical distribution and pattern of Cdhr1 signal always mirrored that of Pcdh15. Taken together, we postulated that the function of Cdhr1 may be to link the OS discs to the CP and that loss of this interaction could represent a novel mechanism of CRD.

### Cdhr1a and pcdh15b interact and form functional connections

To test our hypothesis that cdhr1a and pcdh15b physically interact to connect the OS and CP, we first assessed their interaction using immunoprecipitation (IP). To do so, we cloned cdhr1a cDNA fused

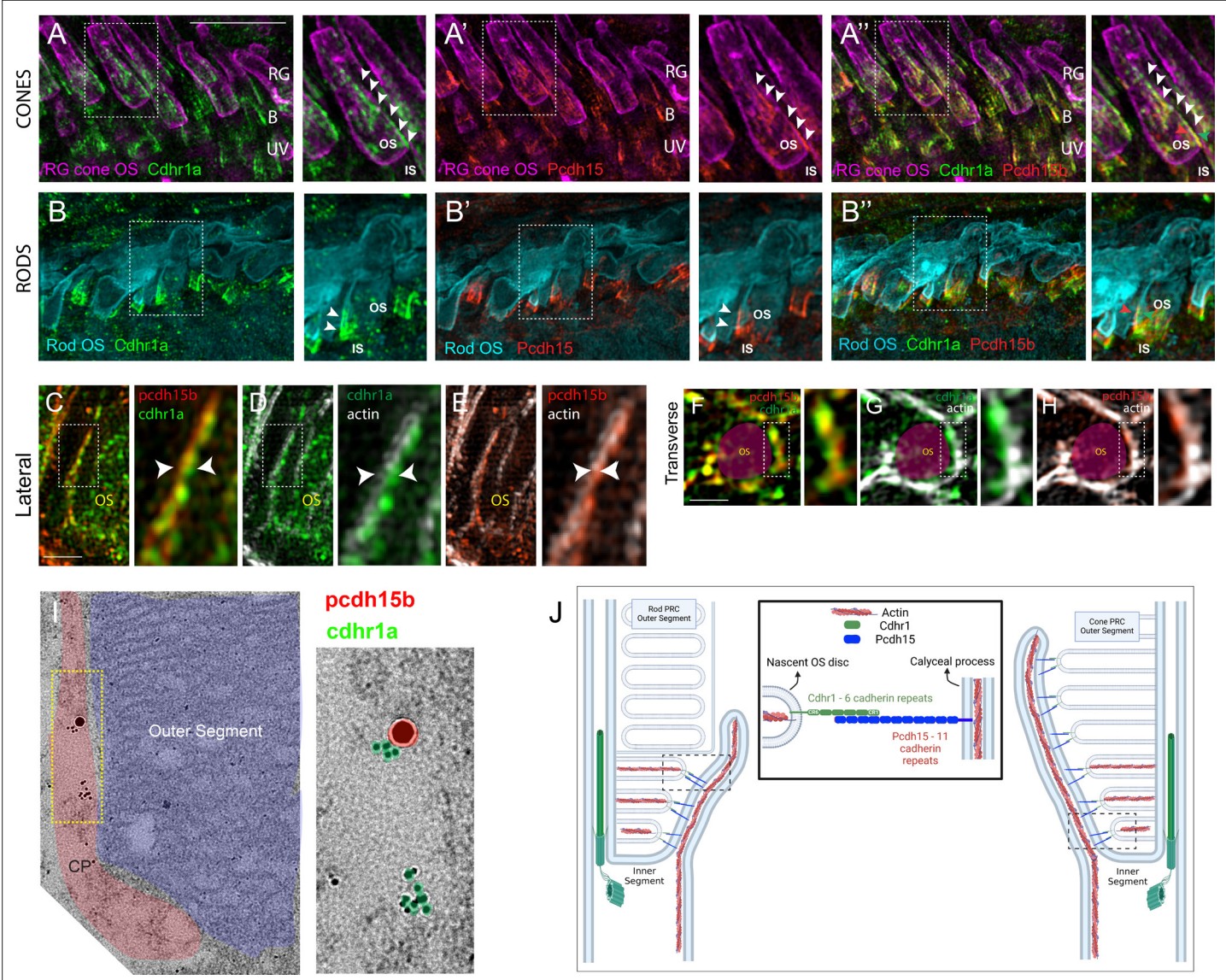

**Figure 1.** Evolutionarily conserved localization of pcdh15 and cdhr1 in photoreceptors predicts interactions linking outer segments and calyceal processes. (**A-A"**) Confocal microscopy of wildtype 5 dpf retinal cryosections probed with cdhr1a antibody (green), Peanut germ agglutinin (PNA - magenta) to label red-green cones, and pcdh15b (red). White boxes indicate the location of the inset enlargement. White arrowheads highlight the linear localization of cdhr1a along cone OSs (**A**) and pcdh15b outlining the calyceal process (**A'**). Merge of all three signals highlights the proximity between cdhr1a and pcdh15b (**A"**). B=blue cones, RG = red/green cones, UV = UV cones. Scale bar = 5 μm. (**B-B"**) Confocal microscopy of wildtype 5 dpf retinal cryosections probed with cdhr1a antibody (green), pcdh15b (red), and wheat germ agglutinin (WGA -teal) to label rods. White boxes indicate the location of the inset enlargement. White arrowheads highlight the linear localization of cdhr1a along rod OSs (**B**) and pcdh15b outlining the calyceal process (**B'**). Merge of all three signals highlights the proximity between cdhr1a and pcdh15b (**B'**). Scale bar = 10 μm. (**C–E**) Structured illumination microscopy (SIM) of 5 dpf wildtype zebrafish retinal cryosections probed with (**C**) cdhr1a (green) and pcdh15b (red), (**D**) pcdh15b (red) and actin (white) and (**E**) pcdh15b (red) and actin (white). White boxes represent the inset enlargement. White arrowheads highlight the juxtaposition or overlap between the cdhr1a, pcdh15b, and actin signals. Scale bar = 2.5 μm. (**F–H**) Structured illumination microscopy (SIM) of wildtype zebrafish whole mount 5 dpf retina in transverse orientation (OS outlined in magenta) probed with (**F**) cdhr1a (green) and pcdh15b (red), (**G**) cdhr1a (green) and actin (white) and (**H**) pcdh15b (red) and actin (white). White boxes represent the inset enlargement. Scale bar = 2.5 μm. (**I**) Immuno-gold-TEM detection of cdhr1a (green) and pcdh15b (red) in 5 dpf wildtype retinal sections. The CP is outlined in red and the OS in blue. Scale bar = 500 nm (**J**) Diagrammatic model of the connection between the OS discs and CPs in both rod and cone cells mediated by the interaction between OS-bound cdhr1a and CP-bound pcdh15b.

The online version of this article includes the following figure supplement(s) for figure 1:

**Figure supplement 1.** Expression of USHR genes in the zebrafish retina.

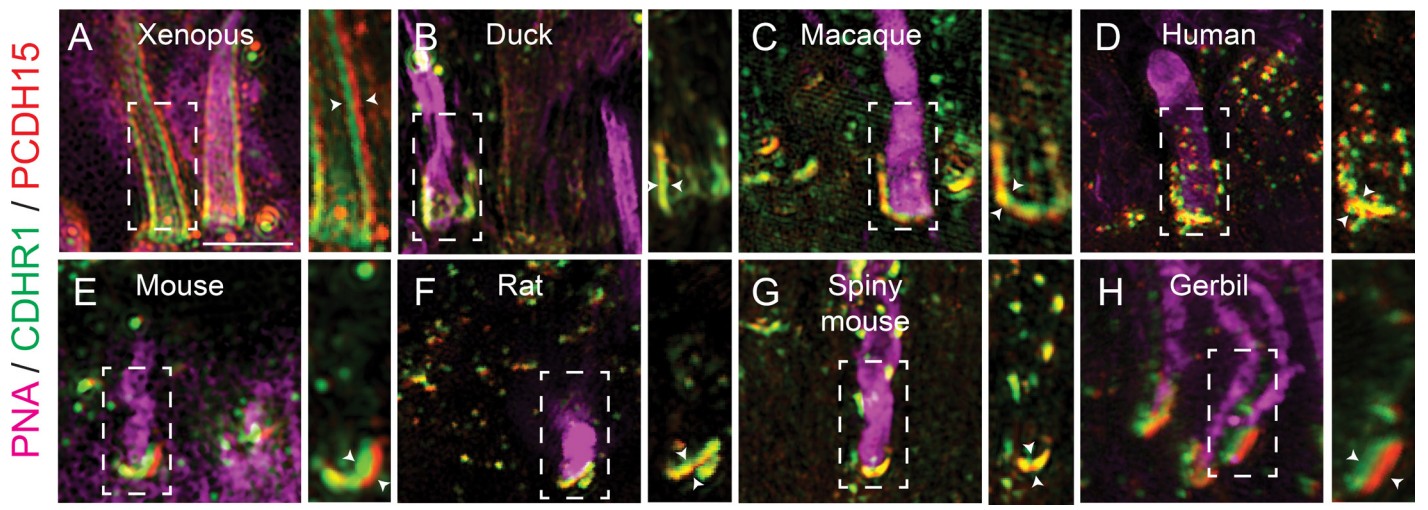

**Figure 2.** Evolutionarily conserved localization of pcdh15 and cdhr1 in photoreceptors predicts interactions linking outer segments and calyceal processes. Structured illumination microscopy (SIM) of wildtype xenopus (**A**), mallard duck (**B**), macaque (**C**), human (**D**), mouse (**E**), rat (**F**), spiny mouse (**G**), and gerbil (**H**), retinal sections probed for cdhr1 (green), pcdh15 (red), and PNA (magenta) to label cone outer segments. White boxes represent the inset enlargements. White arrowheads highlight the juxtaposition of the cdhr1a and pcdh15b signals in each species. Scale bar = 5 µm.

to a C-terminal FLAG tag into a mammalian expression vector and pcdh15b fused to a C-terminal MYC tag. Using HEK293 cells, we performed co-transfection and subsequently FLAG and MYC pull-down assays. Our results show that cdhr1a and pcdh15b can reciprocally pull each other down in the IP assay (*Figure 3A*). While IP can indicate interaction between proteins or between complexes of proteins, our hypothesis predicts that the interactions between cdhr1a and pcdh15b are functional in that they connect the OS and CP in extracellular space. To test whether cdhr1a and pcdh15b can form functional extracellular complexes, we used the K562 leukemia cell line, which lacks any endogenous cadherin protein expression and therefore does not form cell aggregates. Based on this fact, K562 cells are often used to test cadherin-cadherin interactions (*Schreiner and Weiner, 2010*). As such, we transfected K562 cells with either cdhr1a-FLAG or pcdh15b-MYC to test whether their interaction can result in K562 aggregation (*Figure 3B*). Introduction of either cadherin alone resulted in minimal aggregation, either in total number of aggregates or the number of cells per aggregate (*Figure 3C–E*). Conversely, when we mixed the cdhr1a-FLAG and pcdh15-MYC expressing cultures, we observed a significant increase in the total number of aggregates and the number of cells in each aggregate (*Figure 3C–E*). Based on these results, we conclude that cdhr1a and pcdh15b can form cadherin-based junctions in extracellular space and therefore have the potential to establish connections between the OS and CP in the retina.

## Generation of a zebrafish cdhr1a loss of function line for the study of cone-rod dystrophy

CDHR1 deficiency has been modeled using the murine model which recapitulates the human phenotypes associated with CRD (*Yusuf et al., 2021*). However, despite two decades since the model was first generated, the mechanism behind PRC degeneration has yet to be elucidated. While recapitulating the disease phenotypes, the mouse model does exhibit some limitations, including having a rod-rich retina as well as the logistical difficulty in examining numerous timepoints during early development, adolescence, and throughout adulthood. As such, we sought to model CDHR1-mediated CRD using zebrafish, which offers a vertebrate cone-rich model that readily enables detailed molecular examination of PRCs at high temporal resolution. Thanks to high conservation of ocular development, genetics, and function, the zebrafish model has been used to model numerous human ocular diseases including retinitis pigmentosa, LCA, coloboma, and numerous others.

Zebrafish encode two CDHR1 homologs, cdhr1a and cdhr1b, of which only cdhr1a is expressed in the zebrafish photoreceptors (*Piedade et al., 2020*). We targeted cdhr1a using the Alt-R CRISPR technology by designing two crRNA constructs separated by ~170 bp and mapping to exon 6 and exon 7

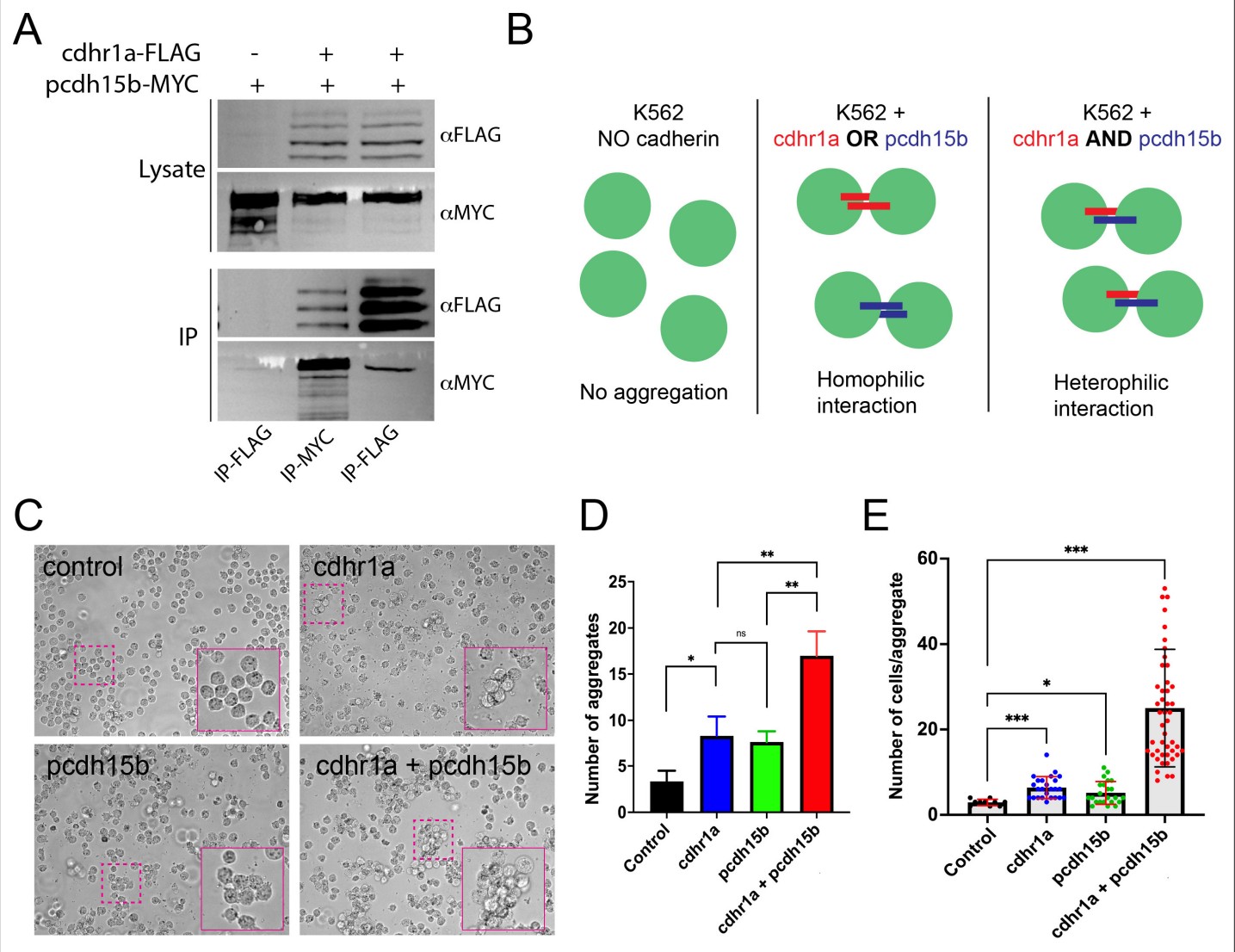

**Figure 3.** Physical interactions between cdhr1a and pcdh15b can facilitate cell-cell adhesion. (**A**) Immunoprecipitation of cdhr1a-FLAG and pcdh15b-MYC performed in HEK293 cell lysates. Pull down of cdhr1a-FLAG using anti-FLAG antibody beads also pulls down pcdh15b-MYC, while pull down of pcdh15b-MYC using anti-MYC antibody beads also pulls down cdhr1a-FLAG. (**B**) Diagrammatic representation of the K562 cell assay for assessing homophilic or heterophilic interactions between cadherins. (**C**) Brightfield microscopy analysis of K562 cell aggregation after transfection with either cdhr1a or pcdh15b or after co-culture of the two transfected populations. Magenta boxes indicate regions enlarged in the insets. (**D**) Quantification of the total number of aggregates observed in a single field of view. (**E**) Quantification of the number of average number of cells in each observed aggregate.

The online version of this article includes the following source data for figure 3:

**Source data 1.** Raw western blot images.

**Source data 2.** Annotated western blot images.

(*Figure 4A*). Upon injection, we generated a stable zebrafish line harboring a 173 bp deletion which results in the excision of the 3' end of exon 6, all of intron 6, and most of exon 7 ultimately generating a premature stop codon at AA146 (*Figure 4B and C*). Cdhr1a protein sequence encodes six extracellular cadherin domains (EC1-6), a transmembrane domain, and a cytoplasmic domain (*Figure 4D*). The 173 bp deletion is predicted to result in a truncation of this polypeptide at AA146, therefore terminating the protein after just the first EC domain. Thus, the remaining protein would lack both the transmembrane and cytoplasmic domain and therefore likely lose all function. The *cdhr1a*[fs146*] line (from here on referred to as *cdhr1a[-/-]*) was bred to homozygosity to create a maternal zygotic mutant population. The maternal zygotic mutants were used for all the subsequent functional experiments.

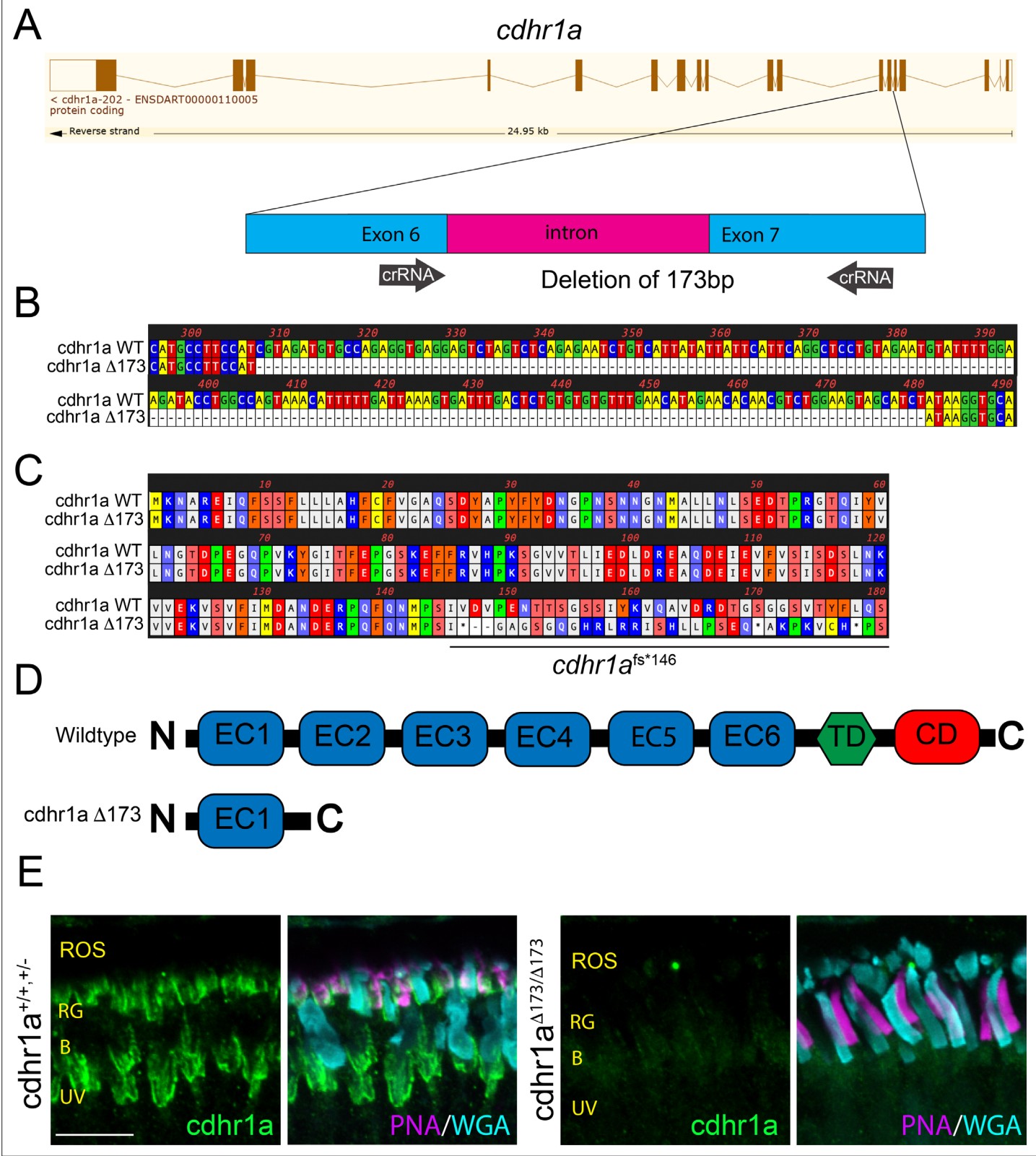

**Figure 4.** Construction and confirmation of the cdhr1a−/− line. (**A**) Exon/intron diagram of cdhr1a. The intron 6-exon 6-intron 7 junction is highlighted and the approximate location of the crRNA (black arrows) is depicted. (**B**) Genomic nucleotide sequence from the cdhr1a^Δ173 line highlighting the sequence location of the 173 bp deletion compared to WT sequence. (**C**) Amino acid sequence alignment between wildtype and the cdhr1a^Δ173 line. Alignment indicates that the 173 bp deletion leads to a frameshift and an immediate premature stop codon at AA146. (**D**) Diagrammatic representation

*Figure 4 continued on next page*

Figure 4 continued

of the domain structure of WT cdhr1a protein vs the cdhr1a$^{\Delta173}$ allele. EC = cadherin domain, TD = transmembrane domain, CD = cytoplasmic domain. (**E**) Confocal microscopy of 5 dpf retinal cryosections from WT and cdhr1a$^{-/-}$ retinas stained with anti-cdhr1a antibody (green), PNA (red-green cones - magenta) and WGA (blue cones - teal). Cdhr1a signal is detected along photoreceptor outer segments in the wildtype and completely missing in cdhr1a$^{-/-}$. ROS = rod outer segment, RIS = rod inner segment. Scale bar = 5 μm.

The online version of this article includes the following figure supplement(s) for figure 4:

**Figure supplement 1.** Expression of *cdhr1a* and *cdhr1b*.

To confirm that our mutant line represented a null allele, we probed for cdhr1a in PRCs using a zebrafish-specific cdhr1a antibody which we previously developed and verified (*Piedade et al., 2020*). As expected, immunohistochemistry (IHC) of 5 dpf wildtype zebrafish retinal cryosections displayed strong cdhr1a staining at the periphery of photoreceptor outer segments (*Figure 4E*). However, cdhr1a$^{-/-}$ samples exhibited a total lack of cdhr1a protein signal, therefore confirming that our mutant line harbors a null allele. Additionally, in situ hybridization data suggests that the allele undergoes nonsense-mediated decay, and we do not detect any retinal expression of cdhr1b as compensation (*Figure 4—figure supplement 1*). Taken together, we were able to generate a novel zebrafish line harboring a heritable null allele in cdhr1a and establish a homozygous population of fish suitable for functional analysis of PRC phenotypes throughout development, adolescence, and adulthood.

## Loss of cdhr1a function in zebrafish results in progressive cone-rod dystrophy

There are no studies to date that have examined effects of CDHR1 loss of function on CPs. As such, to provide a more comprehensive examination of PRC phenotype progression, we examined seven timepoints encompassing early development (5 and 15 dpf), juveniles (30 dpf) as well as adults (90, 180, 360, and 720 dpf). At each timepoint, we first examined the morphology and number of cone cells as it is expected that these cells will be affected first. To visualize cone cell OSs, we used a peripherin2 (prph2) antibody which labels the periphery of OSs in PRCs, including all the cone subtypes. We analyzed five individuals for each genotype and at each time point. All COS in the field of view were measured, which on average included 40 cone cells at early timepoints (5–15 dpf) and 70+ cone cells at later stages (30+ dpf). For consistency, we chose to sample all our timepoints at the central retina (adjacent to the optic nerve). Starting at 5 dpf, we observed that both WT and cdhr1a$^{-/-}$ cone OSs exhibited their expected shape (*Figure 5A and A'*) and displayed smooth vertical peripheral prph2 staining. To determine whether there are any physiological changes to the cone OS (COS), we also measured the OS length based on prph2 signal. When comparing WT to cdhr1a$^{-/-}$ mutants, we observed a slight, but significant increase in the average COS size from 4.79 μm to 5.34 μm (*Figure 5H and I*). At 15 dpf, we again detected an increase in the average COS length in the mutants from 6.74 μm in WT to 7.95 μm (*Figure 5H and I*). We also began to observe that the COS shape appeared to become disorganized, perhaps due to the increase in length (*Figure 5B and B'*). Interestingly, by 30 dpf, when the retina has been fully developed, we observed a significant decrease in average COS length in the mutants, from 7.63 μm in WT to 6.68 μm (*Figure 5H and I*). Additionally, the COS of the mutants continued to appear disorganized, lacking the smooth vertical prph2 labeling observed in WT (*Figure 5C*) and instead displaying an almost disc-like organization (*Figure 5C'*). At 90 dpf, there continued to be a decrease in average COS length compared to WT, 8.40 μm vs 6.81 μm (*Figure 5H and I*) and increased OS disorganization. Furthermore, at 90 dpf, we noted a significant decrease in the number of cones suggesting that there was degeneration of cone cells (*Figure 5D, D' and J*). The most severe phenotypes were observed at the later timepoints. At 180 dpf, COS length further decreased to 4.84 μm while the wildtype remained at 8.62 μm (*Figure 5H and I*). Furthermore, there was a striking reduction in the number of cone cells remaining (*Figure 5J*) while COS morphology exhibited severely stunted and tilted COSs (*Figure 5E'*). Analysis of 360 and 720 dpf timepoints displayed similar results to 180 dpf indicating that a maximum level of cone outer segment degeneration and cone loss is achieved by 180 dpf (*Figure 5F–G'*). Taken together, our examination of COS morphology, size, and number of cone cells indicates that in the absence of cdhr1a function, cone cells lose normal OS morphology which precedes their degeneration, which begins between 1 and 3 months.

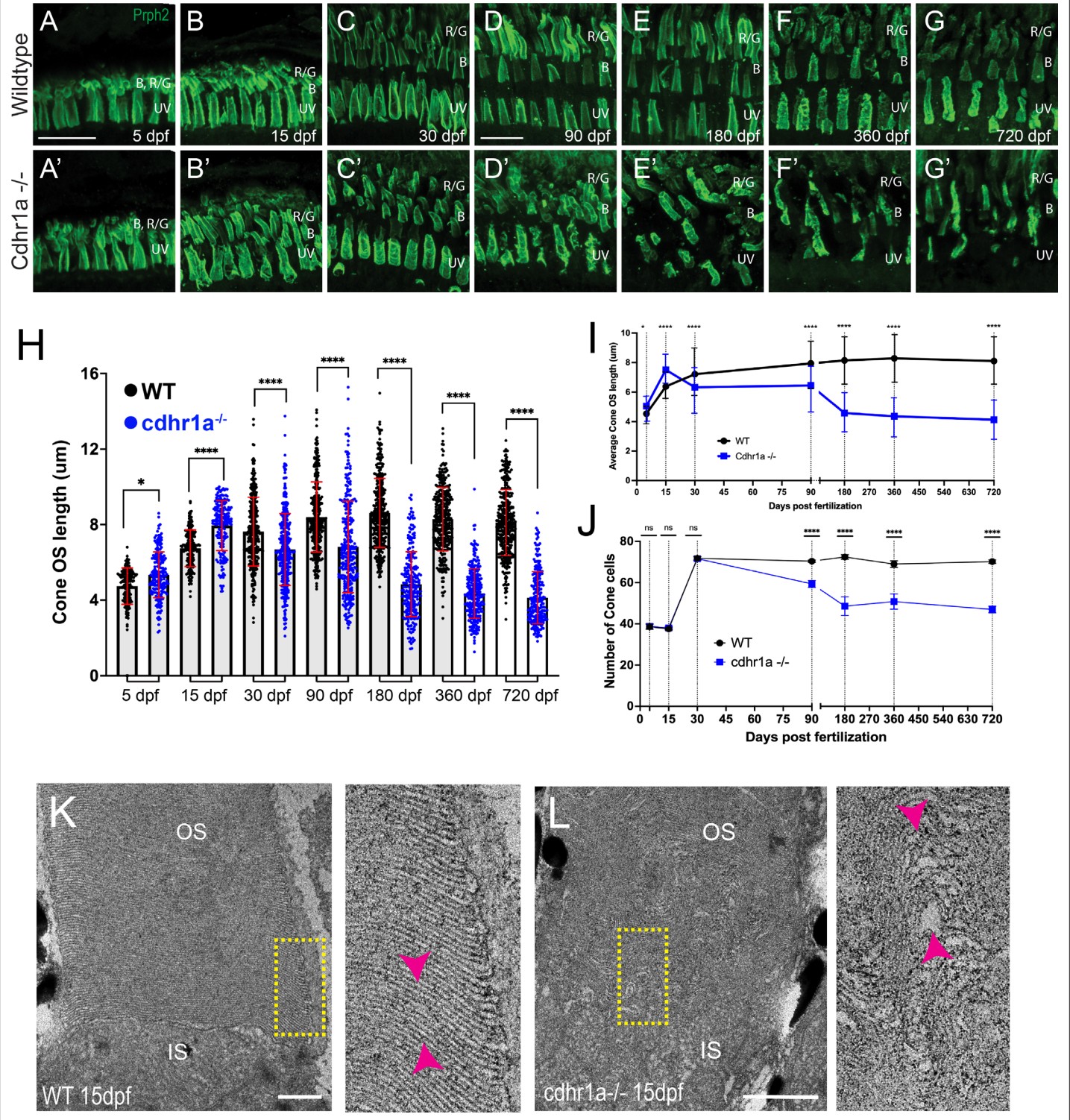

**Figure 5.** Loss of cdhr1a function leads to cone outer segment degeneration. (**A–G**) Confocal microscopy of wildtype retinal cryosections probed with anti-prph2 antibody (green). B=blue cones, RG = red/green cones, UV = UV cones. Scale bar = 10 μm. (**A'-G'**) Confocal microscopy of cdhr1a-/- retinal cryosections probed with anti-prph2 antibody (green). A-B' scale bar = 5 μm, C-G' scale bar = 10 μm. (**H**) Quantification of cone OS length at 5, 15, 30, 90, 180, 360, and 720 dpf measured as length of prhp2 signal in wildtype (black dots) and cdhr1a-/- (blue dots). Standard deviation is shown in red. ****=p < 0.0001. ANOVA = p < 0.0001. (**I**) Line graph depicting the long-term trend of cone OS length changes between wildtype (black) and cdhr1a-/- (blue). (**J**) Line graph depicting changes in the number of cone cells counted in the observation region over time in wildtype (black) compared to cdhr1a (blue). (**K**) Transmission electron microscopy micrographs from a 15 dpf wildtype retina. The yellow rectangle represents the enlarged inset.

*Figure 5 continued on next page*

*Figure 5 continued*

Magenta arrowheads highlight the proper stacking of OS discs. IS = inner segment, OS = outer segment. Scale bar = 500 nm. (**L**) Transmission electron microscopy micrographs from a 15 dpf cdhr1a-/- retina. The yellow rectangle represents the enlarged inset. Magenta arrowheads highlight the improper and distorted stacking of OS discs. IS = inner segment, OS = outer segment. Scale bar = 500 nm.

In addition to fluorescence confocal microscopy analysis of OS morphology, we also examined COSs at electron microscopy resolution using TEM. When comparing WT to cdhr1a mutants at 15 dpf (n=5), we can already detect misorganization of the OS discs in the form of bulges and improperly stacked discs (*Figure 5K–L*). This result corresponds with our fluorescence microscopy data (*Figure 5B'*) where at 15 dpf we already see alteration in the pattern of prph2 staining. Compared to recent work in frogs, we did not observe obvious overgrowth of discs (*Carr et al., 2021*). This may be due to developmental differences between frogs, which undergo metamorphosis, and zebrafish. It may also represent differences in the timepoints analyzed as it remains possible that overgrowth in zebrafish, which exhibits rapid development, may be observed at earlier stages. Taken together, our data indicates that COS integrity is significantly affected by the loss of cdhr1a as early as 15 dpf, and that this results in progressive loss of cone cells starting at 30 dpf, accelerating up to 180 dpf and stabilizing from 360 to 720 dpf (*Figure 5J*).

While cone cells were the primary target of our investigation of CRD in our cdhr1a$^{-/-}$ model, we also examined the fate of rod cells. Like our analysis of cones, we sought to examine rod OS morphology, length, and number of rod cells at the aforementioned timepoints. Due to a delay in the assembly of the rod OSs (ROS) compared to COSs, we used Gnb1 antibody (β subunit of rod specific transducin) to identify rod cells and wheat germ agglutinin (WGA) to visualize their OS in early development (5 and 15 dpf), and subsequently prph2 and WGA at later stages (30–720 dpf). We analyzed five individuals for each genotype and at each time point. All ROS in the field of view were measured and on average, we counted 40 rod cells in each field of view. When analyzing both the morphology and length of ROSs, we observed that WT and cdhr1a$^{-/-}$ ROS were similar in appearance and number at early timepoints (*Figure 6A–C' and H–J*). Interestingly, unlike COSs, ROSs did not exhibit any significant increase in length in the cdhr1a mutants at 15 dpf and 30 dpf (*Figure 6H–I*). The first phenotypic difference for ROS was observed at 90 dpf, where we detected a slight decrease in the average length, 26.6 μm vs 30.3 μm in WT (*Figure 6H and I*). Furthermore, we also noticed the first examples of morphological disorganization (*Figure 6D and D'*) as well as a 3.4% decrease in the number of rod cells (*Figure 6J*). The trend of shorter ROS and disorganized morphology continued up to 180 dpf where the average OS length decreased to 20.4 μm vs 33.7 μm for WT. Furthermore, at 180 dpf, the number of rod cells decreased by 10%, likely stemming from the persistence of disorganized OSs leading to rod cell degeneration (*Figure 6E, E' and H–I*). Similar to what we observed with cones, rod phenotypes were not further exacerbated at either 360 or 720 dpf (Fig F-G', H-J). Taken together, we conclude that rod cells are only marginally affected by the loss of cdhr1a function in early development and early adulthood and show significant phenotypic effects only starting at 180 dpf. Based on our data, we predict that the molecular mechanism driving CRD in our mutant line is primarily involved with the maintenance of COS homeostasis. Rod effects are either secondary responses to the loss of cones or due to differences in the function of cdhr1a between rods and cones.

## Loss of cdhr1a function leads to CP disorganization

Since our hypothesis involves connections between OSs and CPs, we next examined the consequence of cdhr1a function on CPs. Several previous studies have established that zebrafish exhibit robust numbers of CPs whose assembly coincides with formation of the COSs and ROSs (*Sharkova et al., 2024*; *Miles et al., 2021*). To examine CP integrity, we measured their length and observed overall morphology using actin antibody staining which is commonly used to assess CP size and morphology (*Sahly et al., 2012*; *Sharkova et al., 2024*; *Miles et al., 2021*; *Schietroma et al., 2017*). We measured every rod and cone CP observed in the field of view from five individuals at each time point. To assign the correct identity of the CP and correctly measure its length, counterstains to identify cones (PNA) vs rods (gnb1 or WGA) and retinal location using prph2 staining were used. CP measurements were done using prph2 counter staining to delineate the base of the OS as the starting point of the CP (*Figure 7*). Imaging was restricted to sections collected from our standardized anatomical location which included the lens. In line with previous results, we observed a gradual increase in CP length in

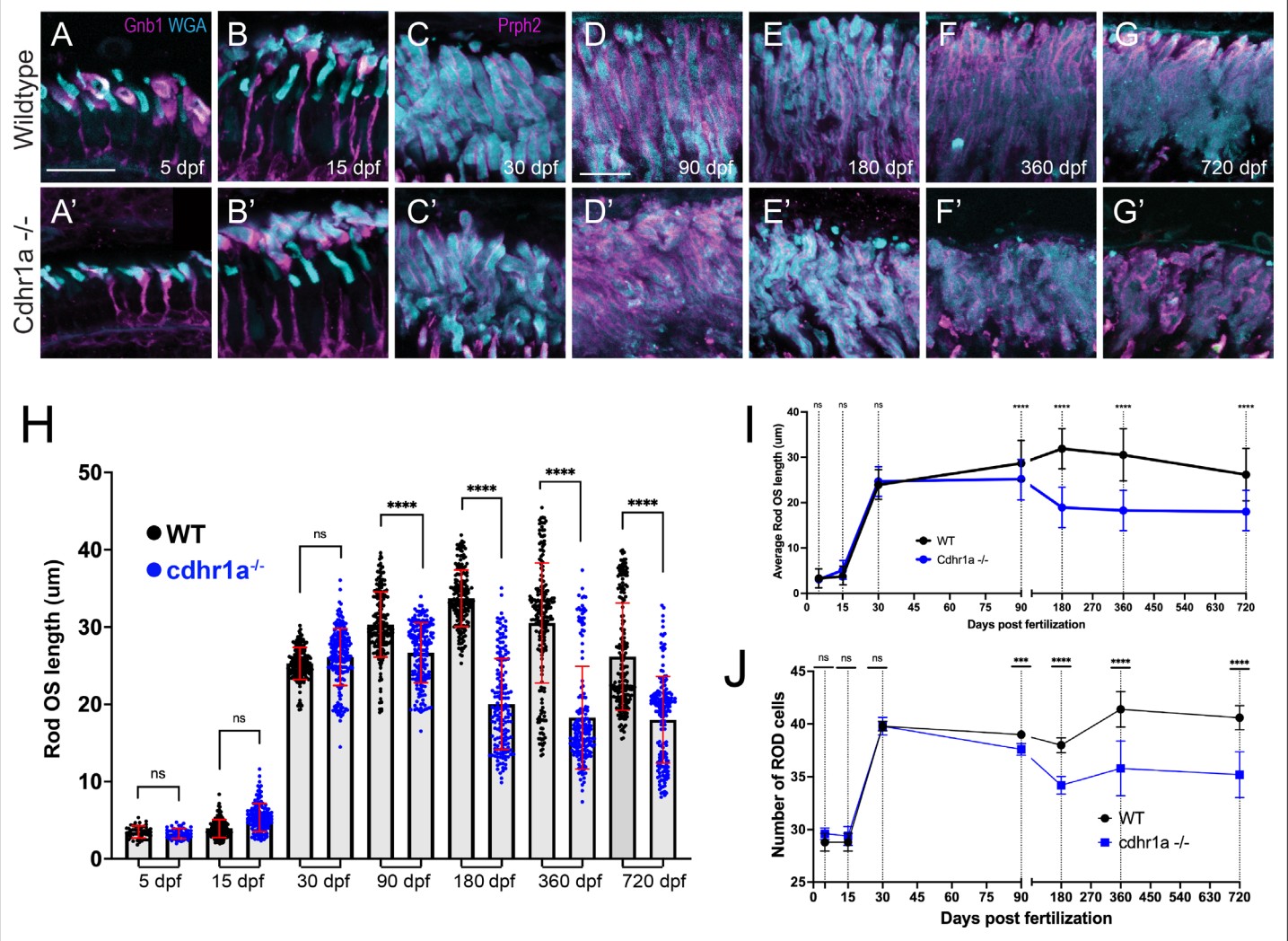

**Figure 6.** Loss of cdhr1a function leads to late-onset rod outer segment degeneration. (**A–B**) Confocal microscopy of wildtype retinal cryosections probed with Gnb1 antibody (magenta) and WGA (teal) to identify rod outer segments at 5 and 15 dpf. Scale bar = 10 µm. (**A'-B'**) Confocal microscopy of cdhr1a[-/-] retinal cryosections probed with Gnb1 antibody (magenta) and WGA (teal) to identify rod outer segments at 5 and 15 dpf. Scale bar = 10 µm. (**C–G**) Confocal microscopy of wildtype retinal cryosections probed with prph2 antibody (magenta) and WGA (teal) to identify rod outer segments. A-B' scale bar = 5 µm, C-G' scale bar = 10 µm. (**C'-G'**) Confocal microscopy of cdhr1a[-/-] retinal cryosections probed with prph2 antibody (magenta) and WGA (teal) to identify rod outer segments. Scale bar = 10 µm. (**H**) Quantification of rod OS length at 5, 15, 30, 90, 180, 360, and 720 dpf measured as length of WGA signal in wildtype (black dots) and cdhr1a[-/-] (blue dots). Standard deviation is shown in red. **=p < 0.001, ****=p < 0.0001. ANOVA = p < 0.0001. (**I**) Line graph depicting the long-term trend of rod OS length changes between wildtype (black) and cdhr1a[-/-] (blue). (**J**) Line graph depicting changes in the number of rod cells counted in the observation region over time in wildtype (black) compared to cdhr1a[-/-] (blue).

both cones and rods throughout early development (5–15 dpf) and a steady state from adolescence to adulthood (30–180 dpf; *Figure 7Q and R*). Furthermore, we observed that zebrafish cone CPs are significantly longer, more than double, than their rod counterparts at all timepoints analyzed. Longer cone CPs were previously also observed in macaque retina (*Sahly et al., 2012*). Next, we compared wildtype CPs to those of the cdhr1a[-/-] mutants. At 5 and 15 dpf, we observed that morphologically cone CPs in cdhr1a[-/-] resembled those of WT (*Figure 7A–B'*). However, even at 5 dpf, measurements of CP length indicated a slight decrease in CP size in the mutants, 3.70 µm vs 3.49 µm (*Figure 7O*), while at 15 dpf, mutant CPs measured longer than wildtype, 5.54 µm vs 6.72 µm (*Figure 7B and B'*). Similar results were observed for rods where mutant CPs were longer at 15 dpf (*Figure 7I–I' and P*). Starting at 30 dpf, we began to observe not only morphological disorganization of actin staining along CPs, but also a significant decrease in overall CP length in cdhr1a[-/-] samples (*Figure 7C and C'*). Cone CPs appear to have lost their tether to the OS and tilt away from the OS. Interestingly,

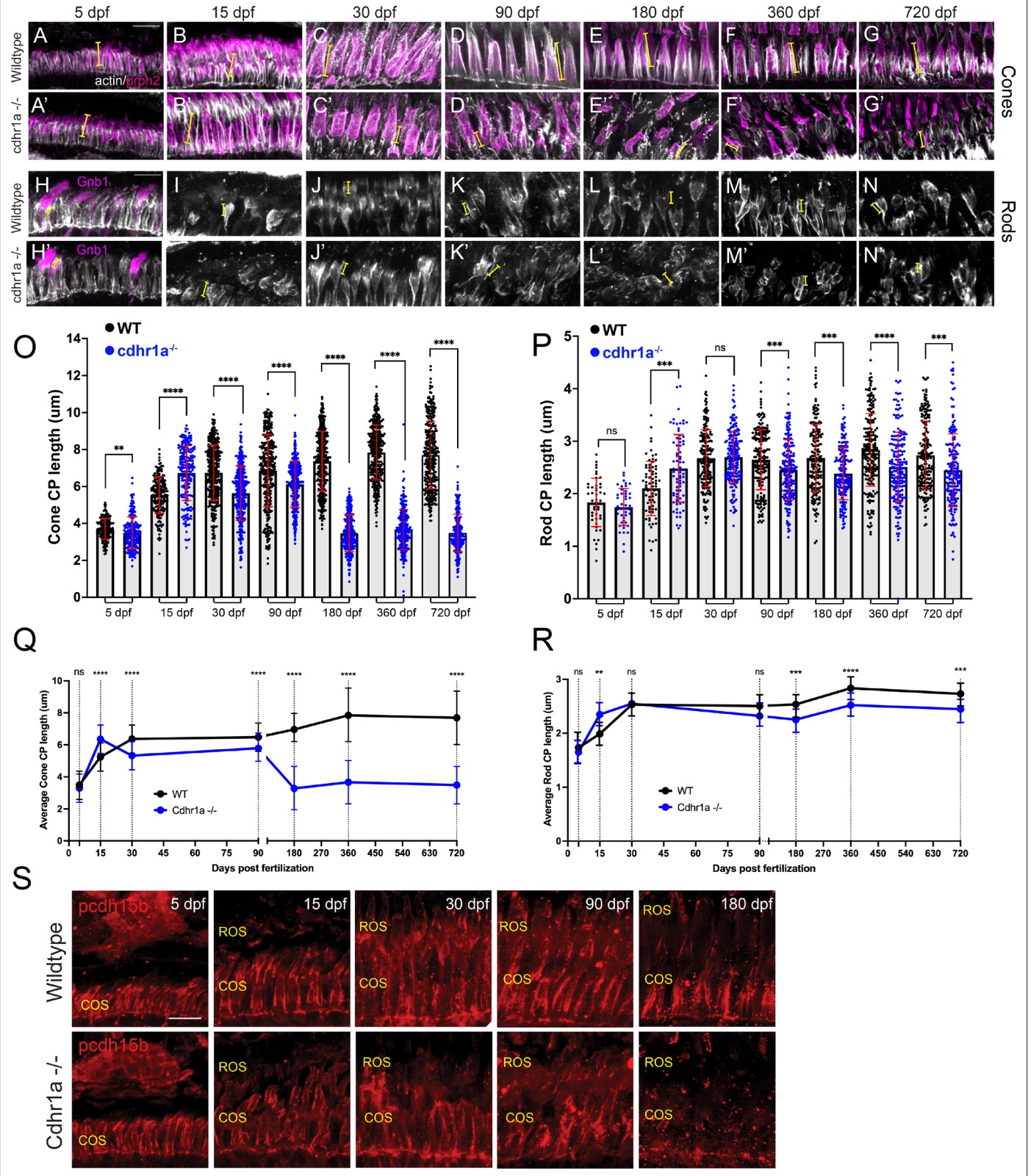

**Figure 7.** Calyceal processes in cones are disorganized in the absence of cdhr1a function. (**A–G**) Confocal microscopy of wildtype retinal cryosections probed with PNA (magenta) and actin-antibodies (white) at various timepoints detecting cone CPs. Typical length of CPs observed is demonstrated by yellow lines. Scale bar = 5 μm. (**A′-G′**) Confocal microscopy of cdhr1a⁻/⁻ retinal cryosections probed with actin-antibodies (white) at various timepoints detecting cone CPs. Typical length of CPs observed is demonstrated in magenta. (**H–N**) Confocal microscopy of wildtype retinal cryosections probed

*Figure 7 continued*

with actin-antibodies (white) and gnb1 (**A'**) (magenta) at various timepoints detecting rod CPs. Typical length of CPs observed is demonstrated in yellow. (**H'-N'**) Confocal microscopy of cdhr1a$^{-/-}$ retinal cryosections probed with actin-antibodies (white) and gnb1 (**F'**) (magenta) at various timepoints detecting rod CPs. Typical length of CPs observed is demonstrated by yellow lines. Scale bar = 5 µm. ROS = cone outer segment layer. (**O**) Quantification of cone cells CP length based on measurements of actin staining in wildtype (black) compared to cdhr1a$^{-/-}$ (blue) at 5, 15, 30, 90, 180, 360, and 720 dpf. Standard deviation is shown in red. **=p < 0.01, ****=p < 0.0001. (**P**) Quantification of rod cells CP length based on measurements of actin staining in wildtype (black) compared to cdhr1a$^{-/-}$ (blue) at 5, 15, 30, 90, 180, 360, and 720 dpf. Standard deviation is shown in red. ns = not significant, ***=p < 0.001, ****=p < 0.0001. (**Q**) Line graph depicting the long-term trend of cone cells CP length changes between wildtype (black) and cdhr1a$^{-/-}$ (blue) over time. **=p < 0.01, ****=p < 0.0001. (**R**) Line graph depicting the long-term trend of rod cells CP length changes between wildtype (black) and cdhr1a$^{-/-}$ (blue) over time. ns = not significant, ***=p < 0.001, ****=p < 0.0001. (**S**) Confocal microscopy of wildtype (top panels) and cdhr1a$^{-/-}$ (bottom panels) retinal cryosections probed with pcdh15b antibody (red) at various timepoints. Scale bar = 5 µm. COS = cone outer segment layer, ROS = rod outer segment layer.

cdhr1a$^{-/-}$ rod CPs do not show any effects in their morphology or length at 30 dpf (*Figure 7H and H'*). This trend continued and intensified in cone CPs at the 90 and 180 dpf timepoints (*Figure 5D–E' and F*). In fact, by 180 dpf, we observed the length of cone CPs shorten to what it was at 5 dpf, which is half of what WT exhibits at 180 dpf, 7.36 µm vs 3.47 µm (*Figure 5E, E' and F*). Like we observed for COS length, later timepoints of 360 or 720 dpf did not demonstrate further deterioration of the cone CPs (*Figure 7F–G', O and Q*). For rod CPs, we observed a much subtler effect, where the CP length isn't affected until 90 dpf, and even at 180 dpf the degree of decrease in size, 2.67 µm vs 2.39 µm, while significant was not relatively as large compared to cone CPs (*Figure 7K–L' and P*). Similar to cone CPs, later timepoints did not exhibit any further decrease in rod CP length or organization (*Figure 7M–N', P and R*). Having noted significant effects on CP size and morphology in the absence of cdhr1a function, we also examined for consequences on pcdh15b localization. At 5 dpf, cdhr1a$^{-/-}$ cone and rod cells displayed wildtype like localization of pcdh15b (*Figure 7O*). However, pcdh15b localization in subsequent timepoints mirrored the disorganization of the CP previously observed using actin (*Figure 7A–E'*). By 180 dpf, very little pcdh15b signal remains (*Figure 7J*), suggesting significant disorganization of CPs but not its complete disassembly. Based on our actin CP staining as well as pcdh15b signal, we conclude that at early timepoints (5–90 dpf), the absence of cdhr1a does not affect the assembly of the CP nor localization of pcdh15b to the CP; however, long-term (180 dpf +) absence of cdhr1a correlates with the eventual loss of pcdh15b signal from the CPs and disorganization of its structure as observed by actin staining.

## Pcdh15b is necessary for proper localization of cdhr1a

Recent studies in zebrafish have indicated that loss of pcdh15b results in malformation of OSs and PRC degeneration (*Miles et al., 2021*). To examine how the loss of pcdh15b compares to the loss of cdhr1a, we generated a CRISPR/Cas9 knockout line by targeting two crRNAs in exon 5, which resulted in a heritable 68 bp deletion (*Figure 8A*). The consequence of the deletion was a frame shift at AA117 and a premature stop at AA118. Like our cdhr1a$^{fs*146}$ allele, the *pcdh15b$^{fs117*118}$* allele (subsequently referred to as pcdh15b$^{-/-}$) results in a truncation of the protein before either the transmembrane domain or the cytoplasmic domain, thus likely representing a null. To confirm the null phenotype, we probed for pcdh15b in retinal cryosections using IHC and observed a lack of pcdh15b signal in pcdh15b$^{-/-}$ samples, validating the allele as a null (*Figure 8B–B''*). Due to the critical function of pcdh15b in the inner ear, pcdh15b$^{-/-}$ larva were only viable up to approximately 10 dpf, an outcome previously reported in zebrafish (*Miles et al., 2021*). To observe the effects of pcdh15b loss of function on PRCs, we first analyzed cone OS morphology and size. Due to the immaturity of rods at 5 and 10 dpf and cone-specific effects observed in the cdhr1a$^{-/-}$ mutant, we focused our attention on cone cells. Using prph2 staining, we observed little effect on COS morphology or length at 5 dpf (*Figure 8C–C''*). However, at 10 dpf, pcdh15b$^{-/-}$ COSs displayed disorganization and a slight decrease in COS length (*Figure 8D–D'' and E*). Next, we examined CP morphology, where at 5 dpf CP morphology in mutants resembled that of wildtype (*Figure 8F–F''*); however, by 10 dpf mutant CPs showed signs of disorganization (*Figure 8G–G''*). Quantification of CP length at 5 dpf pcdh15b$^{-/-}$ larva revealed a slight, yet significant increase in CP length, although by 10 dpf no difference in CP length between wildtype and mutant was detected (*Figure 8H*). Lastly, we examined whether loss of pcdh15b had any effects on cdhr1a localization. To quantify, we measured the length of cdhr1a signal along the OS. In doing so, we observed that loss of pcdh15b did not preclude cdhr1a from localizing to OS; however, as

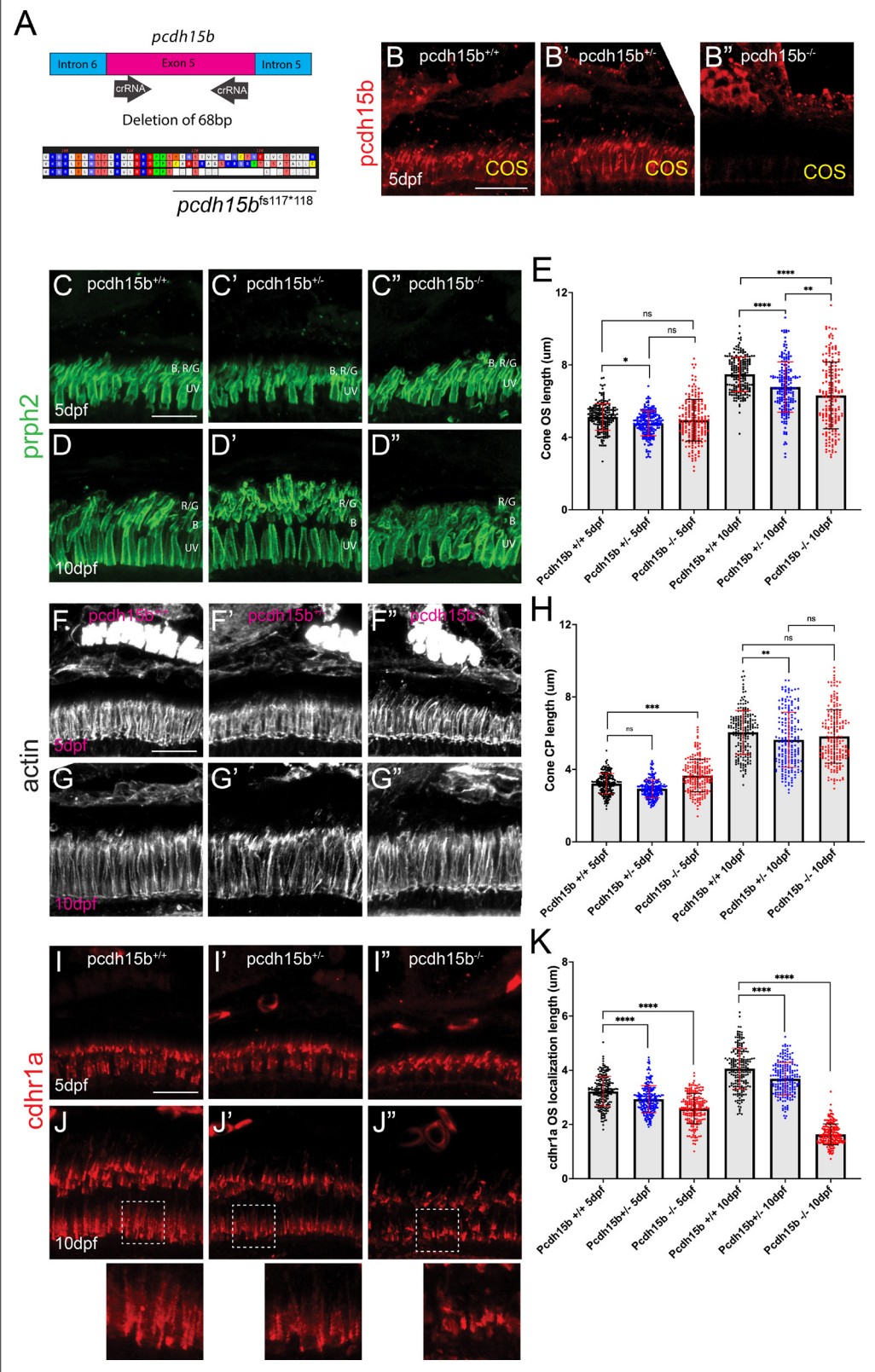

**Figure 8.** Loss of pcdh15b function leads to early cone OS defects and mis-localization of cdhr1a. (**A**) Diagrammatic representation of the CRISPR/Cas9 strategy for generating a pcdh15b loss-of-function allele. crRNAs were targeted to the flanking sequences of exon 5 and resulted in a heritable deletion of 68 bp which resulted in a frameshift (fs) at AA117 and a premature stop codon (*) at AA118. (**B-B"**) Confocal microscopy of

*Figure 8 continued on next page*

*Figure 8 continued*

retinal cryosections from wildtype, pcdh15b$^{+/-}$, and pcdh15b$^{-/-}$ individuals probed with pcdh15b antibody (red). Scale bar = 10 μm.(**C-D″**) Confocal microscopy of retinal cryosections from wildtype, pcdh15b$^{+/-}$, and pcdh15b$^{-/-}$ individuals probed with prhp2 antibody (green) to visualize the cone outer segments (COS) at 5 dpf (**C-C″**) and 10 dpf (**D-D″**). B=blue cones, RG = red/green cones, UV = UV cones. Scale bar = 10 μm. (**E**) Quantification of COS length using prph2 signal for each genotype at each time point is depicted on the right. Standard deviation is shown in red. ns = not significant, **=p < 0.01. ****=p > 0.0001. ANOVA = p < 0.0001. (**F-G″**) Confocal microscopy of retinal cryosections from wildtype, pcdh15b$^{+/-}$, and pcdh15b$^{-/-}$ individuals probed with actin antibody (white) to visualize cone CPs at 5 dpf (**F-F″**) and 10 dpf (**G-G″**). Scale bar = 10 μm. (**H**) Quantification of CP length using actin signal for each genotype at each time point is depicted on the right. Standard deviation is shown in red. ns = not significant, **=p < 0.01. ****=p > 0.0001. ANOVA = p < 0.0001. (**I-J″**) Confocal microscopy of retinal cryosections from wildtype, pcdh15b$^{+/-}$, and pcdh15b$^{-/-}$ individuals probed with cdhr1a antibody (red) at 5 dpf (**I-I″**) and 10 dpf (**J-J″**). White boxes indicate regions enlarged. Scale bar = 10 μm. (**K**) Quantification of cdhr1a OS length for each genotype at each time point is depicted on the right. Standard deviation is shown in red. ns = not significant, ***=p < 0.001. ****=p > 0.0001. ANOVA = p < 0.0001.

early as 5 dpf, pcdh15b mutants displayed a significant decrease in the spread of cdhr1a signal along the OS (*Figure 8I–I″*), which became much more pronounced by 10 dpf (*Figure 8J–J″*). Quantification of cdhr1a signal length along the OS revealed a significant decrease as early as 5 dpf, 3.22 μm vs 2.59 μm, and a major decrease by 10 dpf, 4.06 μm vs 1.64 μm (*Figure 8K*). Taken together, we conclude that while pcdh15b is not required for cdhr1a localization to the OS, spread of cdhr1a signal along the cone OS, which likely represents OS-CP interactions, requires pcdh15b.

## Loss of cdhr1a and pcdh15b synergizes, resulting in severe CP disruption and cone outer segment malformation

Having observed phenotypes affecting CPs and OSs in both cdhr1a and pcdh15b individual mutant lines, we next asked whether a combination of both mutations would lead to more severe outcomes. To do so, we established a *cdhr1a$^{-/-}$; pcdh15b$^{+/-}$* line. By crossing this line, we were able to collect 5 and 10 dpf embryos that lacked both cdhr1a and pcdh15b (cdhr1a$^{-/-}$; pcdh15b$^{-/-}$) function. We then again performed our morphological analysis of cone OSs using prph2 antibody staining and CPs using actin antibody staining. At 5 dpf, we observed little change in COS morphology when comparing cdhr1a$^{-/-}$ to pcd15b$^{-/-}$ mutants (*Figure 9A and B*). COS length, however, differed, where cdhr1a$^{-/-}$ individuals display significantly longer COSs at 5.63 μm versus 4.96 for pcdh15b$^{-/-}$. Interestingly, when comparing the double mutant to either single mutant or any of the other genetic combinations, we observed a significant decrease in COS length, 3.68 μm, as well as disorganized COS morphology (*Figure 9A–D*). This trend was even more pronounced at the 10 dpf timepoint, where disorganization of the COS was highly evident in the double mutant (*Figure 9E–H*). Furthermore, the double mutant COS length decreased to 4.31 μm compared to 7.83 μm for cdhr1a$^{-/-}$ and 6.31 μm for pcdh15b$^{-/-}$ (*Figure 9E–I*). Thus, it appears that concurrent loss of cdhr1a and pcdh15b combines to result in significant decreases in COS length as well as very early morphological COS disorganization.

When examining CP length and morphology, we observed similar trends. At 5 dpf, cdhr1a mutants exhibit elongated CPs with normal morphology, compared to slightly truncated but morphologically normal CPs in pcdh15b mutants (*Figure 9K and L*). Double mutants display normal CP morphology at 5 dpf, albeit their CPs are significantly shorter than wildtype (3.22 μm vs 2.45 μm; *Figure 9N and S*). By 10 dpf, both CP morphology and length are significantly affected in the double mutant compared to either single mutant (*Figure 9O–P and T*). CPs appear severely disorganized, lacking the expected vertical actin staining, instead appearing shorter and bent or tilted. When comparing CP lengths, the double mutant has severely truncated CPs, 2.93 μm vs 6.34 μm in cdhr1a mutants and 5.82 μm in pcdh15b mutants. Taken together, we conclude that the combination of pcdh15b and cdhr1a loss of function enhances both COS disorganization and decreases COS length along with CP morphological disruption and severe truncation even at very early timepoints. These findings further support our hypothesis where cdhr1a functions to link cone OSs with CPs by interacting with pcdh15b.

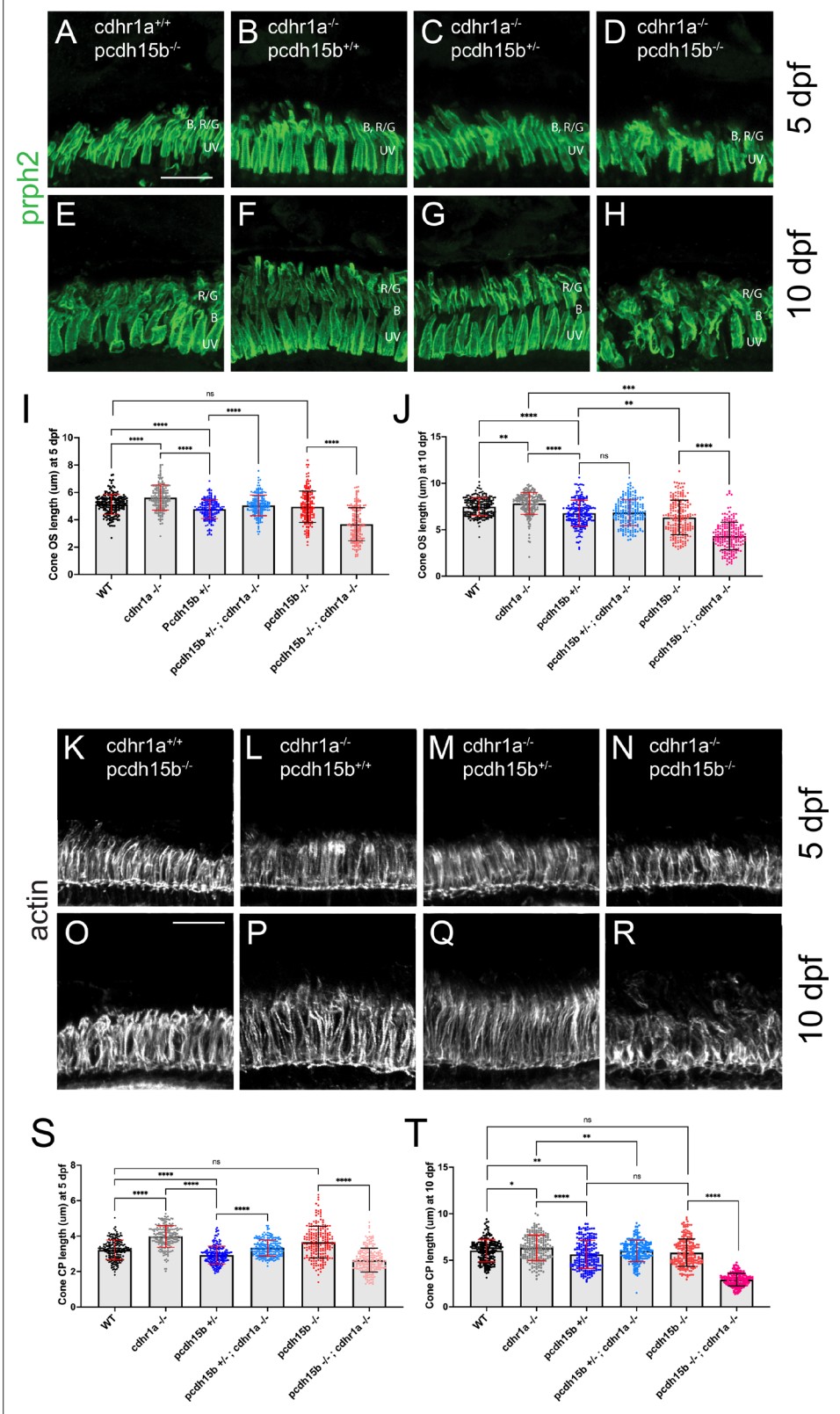

**Figure 9.** Cone phenotypes are exacerbated by simultaneous loss of pcdh15b and cdhr1a. (**A–D**) Confocal microscopy of retinal cryosections from 5 dpf pcdh15b⁻/⁻, cdhr1a⁻/⁻, cdhr1a⁻/⁻; pcdh15b⁺/⁻ and cdhr1a⁻/⁻; pcdh15b⁻/⁻ larva probed with prph2 antibodies (green). COS = cone outer segment. B=blue cones, RG = red/green cones, UV = UV cones. Scale bar = 10 µm. (**E–H**) Confocal microscopy of retinal cryosections from 10 dpf pcdh15b⁻/⁻,

*Figure 9 continued on next page*

*Figure 9 continued*

cdhr1a$^{-/-}$, cdhr1a$^{-/-}$; pcdh15b$^{+/-}$ and cdhr1a$^{-/-}$; pcdh15b $^{-/-}$ larva probed with prph2 antibodies (green). COS = cone outer segment. B=blue cones, RG = red/green cones, UV = UV cones. Scale bar = 10 μm. (**I**) Quantification of cone outer segment length at 5 dpf based on prph2 signal. ns = not significant, ****=p < 0.0001. ANOVA = p < 0.0001. (**J**) Quantification of cone outer segment length at 10 dpf based on prph2 signal. ns = not significant, **=p < 0.01, ****=p < 0.0001. ANOVA = p < 0.0001. (**K–N**) Confocal microscopy of retinal cryosections from 5 dpf pcdh15b$^{-/-}$, cdhr1a$^{-/-}$, cdhr1a$^{-/-}$; pcdh15b$^{+/-}$ and cdhr1a$^{-/-}$; pcdh15b $^{-/-}$ larva probed with actin antibodies (white). COS = cone outer segment. Scale bar = 10 μm. (**O–R**) Confocal microscopy of retinal cryosections from 10 dpf pcdh15b$^{-/-}$, cdhr1a$^{-/-}$, cdhr1a$^{-/-}$; pcdh15b$^{+/-}$ and cdhr1a$^{-/-}$; pcdh15b$^{-/-}$larva probed with actin antibodies (white). COS = cone outer segment. Scale bar = 10 μm. (**R**) Quantification of cone CP length at 5 dpf based on prph2 signal. ns = not significant, ****=p < 0.0001. ANOVA = p < 0.0001. (**T**) Quantification of cone CP length at 10 dpf based on prph2 signal. ns = not significant, *=p < 0.05 **=p < 0.01****,=p < 0.0001. ANOVA = p < 0.0001.

## Discussion

Discovered over 20 years ago, the retinal cadherin CDHR1 is clinically associated with recessive CRD, yet its exact molecular function continues to remain a mystery. In our current study, we found that the zebrafish homologue of CDHR1, cdhr1a, localizes to cone and rod OSs in a pattern mirroring that of the inner segment CP and its own resident cadherin pcdh15b (PCDH15 homolog in zebrafish) (*Figure 1*). Interestingly, the pattern of CDHR1/PCDH15 localization was observed across various species, including amphibians, rodents, birds, primates, and humans, thus indicating a high degree of mechanistic evolutionary conservation (*Figure 2*). Aquatic species, zebrafish, and xenopus, appear to have retained significant length of their cone CPs, particularly along the UV cones, as evident by the length of cdhr1/pcdh15 signal (*Figure 1C and E*). Conversely, in rodents, which have been previously shown to either lack CPs or contain a single vestigial structure, CDHR1/PCDH15 juxtaposition is still maintained, albeit in a very short segment (*Sahly et al., 2012*). Intriguingly, the size of the CP may be related to the diurnal nature of an organism as the gerbil exhibited longer structures compared to rats and mice (*Figure 2H*). Birds and primates exhibit CP lengths longer than rodents but still shorter than fish or frogs (*Figure 1K–L*). In addition to localization, we showed that cdhr1a physically interacts with pcdh15b in vitro, via co-immunoprecipitation and in vivo via the K562 cell aggregation assay (*Figure 3*). Furthermore, using immuno-gold TEM, we observed direct interaction between cdhr1a and pcdh15b at the photoreceptor-CP interphase region. Taken together, the localization and interaction findings provide the first potential clue to the molecular function of CDHR1 and the etiology of CDHR1-associated CRD. Based on these findings, we therefore hypothesized that cdhr1a links the OSs with the CPs via interactions with pcdh15b and postulate the absence of this interaction as a mechanism driving cone cell degeneration.

To test our hypothesis, we tracked CRD progression in zebrafish using single cdhr1a$^{-/-}$ or pcdh15b$^{-/-}$ mutant lines as well as a cdhr1a$^{-/-}$, pcdh15b$^{-/-}$ double mutants. Taking advantage of the zebrafish system, we were able to examine CRD progression at numerous developmental and adult stages at the molecular level with high resolution. When examining the cdhr1a$^{-/-}$ model, we documented progressive disorganization and shortening of cone OS starting at 30 dpf, while observing only minor effects in rods (*Figures 5 and 6*). Interestingly, disorganization and shortening of cone OSs at 30 dpf was preceded by slight but significant lengthening at 5 and 10 dpf. This suggests that early development of the OS may be regulated by CP-OS interactions to ensure proper rate of OS growth (*Figure 5H*). Cdhr1a$^{-/-}$ cones exhibited shortened and disorganized CPs as early as 5 dpf, indicating that defects in CP-OS interactions precede and may therefore lead to cone OS instability (*Figure 7*). However, we also observed mutant CP overgrowth at 15 dpf, suggesting that CP assembly or maintenance is misregulated in the absence of cdhr1a (*Figure 7O*). A striking phenotype observed in mutant cone OSs was the misorganization of prph2 signal. Normally, prph2 signal appears smoothly along the periphery of cone OS and largely absent from the center; however, in cdhr1a$^{-/-}$ cones, we observe an almost disc-like horizontal staining of prph2 throughout the OS as early as 15 dpf and very clearly by 30 dpf (*Figure 5B′–G′*). Prph2 misorganization was also accompanied by the appearance of cone OS bending and tilting, compared to wildtype (*Figure 5A–G′*). Similar prph2 localization phenotypes have been observed in prom1b mutant zebrafish, which is interesting as prom1 has been shown to interact with Cdhr1 (*Yang et al., 2008*). Perhaps loss of prom1b results in the mis-localization or mis-regulation of cdhr1a in the cone OS and therefore also compromises CP-OS interactions leading

to similar phenotypes. Finally, using TEM, we observed that cdhr1a$^{-/-}$ cone OSs displayed disorganized disc stacking, which supports the observed prph2 localization phenotype (*Figure 5K and L*). Compared to previous studies of cdhr1 function in frogs or pcdh15b function in zebrafish, we did not observe disc overgrowth in 15 dpf zebrafish. This may result from disorganization and loss of OS integrity already by 15 dpf. Disc overgrowth may therefore be observed at earlier timepoints but those were not examined in our studies. Several previous studies have postulated that CP-OS interactions are critical for stability of cone OSs (*Sahly et al., 2012*; *Sharkova et al., 2024*; *Schietroma et al., 2017*). Our data supports this notion where cdhr1a mutant cone OSs are shorter and often appear tilted to one side. Similar OS phenotype results were also observed in pcdh15b mutants up to 10 dpf which is in line with previous reports from zebrafish and what was also observed in xenopus embryos injected with Pcdh15 morpholinos (*Miles et al., 2021*; *Schietroma et al., 2017*). Surprisingly, pcdh15b heterozygotes exhibited slight but significant reduction in cone OS and CP length (*Figure 8E and H*). This suggests that steady-state levels of pcdh15b protein are critical for its function. Pcdh15b mutants display cone OS defects earlier than cdhr1a$^{-/-}$ and do not exhibit the early increase in OS length (*Figure 8*). This might suggest that the absence of pcdh15b interferes with CP function, while the absence of cdhr1a interferes with OS-CP connections but not the CP itself. In humans, PCDH15 mutations are associated with rod-cone dystrophy; however, due to the lethality of the pcdh15b mutation by 15 dpf, we are not able to compare long-term pcdh15b mutant phenotypes with cdhr1a mutants. In the double mutants, due to the absence of both cadherins, we observed shorter and disorganized CPs and cone OSs as early as 5 dpf, normally not seen in either single mutant alone (*Figure 9*). By 10 dpf, the double mutant phenotypes were severe and comparable to what we observed in cdhr1a$^{-/-}$ mutants by 30+ dpf (*Figure 9*). The exacerbation of phenotypes in the double mutant further supports our model where cdhr1a and pcdh15b function to anchor cone OSs with CPs. Interestingly, these results also suggest that in the absence of either cdhr1a or pcdh15b, there may be redundant mechanisms that aid in OS-CP anchoring. Perhaps other retinal cadherins can interact with cdhr1a or pcdh15b, albeit not as strongly or efficiently, and delay the early phenotypes until the OSs are fully mature. Taken together, our data along with previous studies strongly supports the hypothesis that in the absence of cdhr1a function, the CP-OS interaction is compromised and thus hampers cone OS integrity.

While the focus of our study was on cone cells, we also characterized effects of cdhr1a loss on rods. Rod OSs, nor their CPs appear to be significantly affected by the loss of cdhr1a function until later stages, 3 months or longer (*Figures 6 and 7*). Albeit significant, these effects were much less obvious compared to the effects on cones, as one might expect for a CRD model. Furthermore, while cone CPs displayed gross morphological defects, rod CPs were only slightly affected even at later timepoints (*Figures 6 and 7*). This likely stems from the fact that rod CPs are inherently much smaller than cone CPs and may not serve as critical a role in rods as we predict they do in cones (*Sahly et al., 2012*; *Sharkova et al., 2024*; *Miles et al., 2021*; *Schietroma et al., 2017*). Comparing rod CP length over the course of development and adulthood in our study, we note that rod CPs are on average approximately one third the size of cone CPs (*Figure 7*). Furthermore, compared to the length of rod OS, rod CPs encompass a very small portion of the OS, while cone CPs extend up to three quarters of the cone OS length. Based on these observations of rod CPs, we hypothesize that zebrafish rod CPs only extend along the newly forming OS discs and do not provide structural support to the ROS. As such, we support the hypothesis postulated by *Burgoyne et al., 2015* that in rods, cdhr1a functions to anchor newly formed OS discs to facilitate proper maturation of the discs prior to release into the OS. While this differs from its function in cones, cdhr1a still likely relies on its interaction with pcdh15b to serve this function as the two proteins clearly juxtapose in rod cells across all the species we examined, including rodents which have been hypothesized to lack CPs or have very simplified versions of CPs compared to other species (*Sahly et al., 2012*).

The discovery of the interaction between cdhr1a and pcdh15b is novel. Previous studies have suggested that PCDH15 and CDH23 are binding partners in the retina as they are well known to interact in the inner ear stereocilia (*Schietroma et al., 2017*). However, our work, and others, have shown that cdh23 is not expressed in zebrafish retina, nor does loss of cdh23 result in ocular phenotypes (*Figure 1—figure supplement 1*). Work from macaque retina shows cdh23 expression in the OS/IS region but does not display the prototypical CP staining as observed with pcdh15 or actin (*Sahly et al., 2012*). While several Usher 1 proteins other than pcdh15 have been found to localize to

the CP, including harmonin, myosin, and sans, there has been a missing link between pcdh15 in CP and its potential partner the OS, which we now propose to be Cdhr1. In our study, cdhr1a-pcdh15b interactions are supported by immunoprecipitation and cell aggregation assays and by the fact both are cadherin proteins that can facilitate heterophilic interactions (*Figure 3*). PCDH15 is known to generate cis dimers like cdh23, and as such the cdhr1a-pcdh15b interaction may in fact involve hetero-tetrameric complexes (*Elferich et al., 2021*). Additional evidence for the cdhr1a-pcdh15b interaction was the observation that cdhr1a localization along the cone OS was significantly impaired in the absence of pcdh15b function. At 5 dpf and even more so at 10 dpf, the length of cdhr1a signal along the cone OS was severely reduced, while cone OS length was only slightly affected (*Figure 9*). Thus, we conclude that while pcdh15b is not necessary for cdhr1a to localize to the base of cone OSs, its absence prevents cdhr1a from forming connections to the CP and subsequently extending along the length of the OS. Other proteins may further regulate cdhr1a-pcdh15b complexes such as prom1b, which is known to interact with cdhr1, as well as other Usher 1 proteins found in the CP. CDHR1 is also known to undergo proteolytic cleavage to remove its extracellular cadherin domains, which could further regulate the interaction and stability of the CP-OS attachments (*Rattner et al., 2004*). It remains unclear if the cleavage occurs in both rod and cone cells. We hypothesize that proteolytic cleavage is specific to rods, where this would aid in regulating the release of the newly matured OS discs, while cleavage would be absent in cone cells to ensure CP-mediated OS structure and integrity. Interestingly, we have also previously reported that cdhr1a can be targeted by the Siah1 ubiquitin ligase for proteasomal degradation, adding yet another layer to the potential complexity of regulating cdhr1a (*Piedade and Famulski, 2021*). Imaging of siah1 indicates that it also localizes to the OS in a pattern similar to that of cdhr1a (data not shown). Future studies will be required to elucidate the entire complex responsible for the anchoring and maintaining of CPs and OSs in rods versus cones.

Overall, our cdhr1a$^{-/-}$ model corroborates findings from the mouse CDHR1 KO model with early phenotypes found exclusively in cone cells, and only at later stages, 6+months, displaying rod cell OS disorganization and degeneration. In conclusion, we propose that cdhr1a and pcdh15b function to link the cone OS and the CP to maintain cone homeostasis in an evolutionarily conserved manner. Disruption of this interaction may therefore be a potential driving factor in the progression of CRD observed in CDHR1 patients. Future studies will need to focus on the molecular consequences of OS-CP interaction and the identity of the entire repertoire of proteins involved as well as any potential signaling pathways that activate in response to sensing or monitoring OS-CP interactions.

# Materials and methods

**Key resources table**

| Reagent type (species) or resource | Designation | Source or reference | Identifiers | Additional information |
|---|---|---|---|---|
| Gene (*D. rerio*) | cdhr1a | GenBank | BX855592.19 | |
| Gene (*D. rerio*) | pcdh15b | GenBank | CR925795.5 | |
| Strain, strain background *E. coli* | JM109 | Promega | L2005 | |
| Genetic reagent (*D. rerio*) | Cdhr1a-/- | This paper | Cdhr1a$^{fs*146}$ | Deletion of 173 bp (*Figure 4*) |
| Genetic reagent (*D. rerio*) | Pcdh15b-/- | This paper | pcdh15b$^{fs117*118}$ | Deletion of 68 bp (*Figure 8*) |
| Cell line (*H. sapiens*) | Human leukemia cell line | ATTC | K562, CLL-243 | |
| Cell line (*H. sapiens*) | Human embryonic kidney cell line | ATCC | CRL-3216 | |
| Sequence-based reagent (*D. rerio*) | crRNA targeting cdhr1a coding sequence | This paper (Materials and methods) | AB, AE | AB: GTCTGGAAGTAGCATCTATA AE: TCTGGCACATCTACGATGGA |

*Continued on next page*

*Continued*

| Reagent type (species) or resource | Designation | Source or reference | Identifiers | Additional information |
|---|---|---|---|---|
| Sequence-based reagent (*D. rerio*) | crRNA targeting pcdh15b coding sequence | This paper (Materials and methods) | PCDH15B.1.AV PCDH15B.1.AA | AV: CACCACAATGGACTGGATGT AA: CGACTATCCGCACCTCGTGT |
| Antibody | Rabbit α-cdhr1a, (*D. rerio*) Rabbit polyclonal | BosterBio | DZ07988 | (1/100) |
| Antibody | Rabbit a-Peripheirn-2 Rabbit polyclonal | Protein-Tech | 18109–1-AP | (1/100) |
| Antibody | Sheep a-hPCDH15 | Thermo Fisher | PA5-47865 | (1/75) |
| Antibody | Rabbit a-hCDHR1 Rabbit polyclonal | Thermo Fisher | PA5-57832 | (1/100) |
| Antibody | Mouse anti-actin | Thermo Fisher | MA1-140 | (1/100) |
| Antibody | Rabbit anti-GNB1 Rabbit polyclonal | Thermo Fisher | PA5-30046 | (1/100) |
| Antibody | Mouse anti-zfrhodopsin Mouse monoclonal | Fadool lab, Florida State University, FL | 1D1 | (1/100) |
| Antibody | Mouse anti-FLAG Mouse monoclonal | Sigma-Aldrich | F1804 | (1/1000) |
| Antibody | Mouse anti-MYC Mouse monoclonal | Thermo Fisher | MA121316 | (1/1000) |
| Antibody | Goat anti-Rabbit Alexa Fluor 488–5 nm colloidal gold | Thermo Fisher | A-31565 | (1/1000) |
| Antibody | Donkey-Anti-Sheep 25 nm colloidal gold | Electron Microscopy Sciences | 25829 | (1/100) |
| Recombinant DNA reagent (*D. rerio*) | cdhr1a-FLAG (plasmid) | This paper | pCIG2- cdhr1a-FLAG | (Materials and methods) |
| Recombinant DNA reagent (*D. rerio*) | pcdh15b-MYC (plasmid) | This paper | pCDNA3-pcdh15b-MYC | (Materials and methods) |
| Sequence-based reagent | Kpn1-Pcdh15b PCR primer | This paper | PCR primer | (Materials and methods) |
| Sequence-based reagent | Pcdh15b-XhoI PCR primer | This paper | PCR primer | (Materials and methods) |
| Peptide, recombinant protein | Alt-R Cas9 v.3 enzyme | IDTDNA | 1081058 | |
| Commercial assay or kit | Anti-FLAG magnetic beads | Thermo Fisher | A36797 | |
| Commercial assay or kit | Anti-MYC magnetic beads | Thermo Fisher | 88842 | |
| Other | Peanut Germ Agglutinin conjugated with a 488 fluorophore | Biotium | #29060 | (1/100) |
| Other | Wheat Germ Agglutinin conjugated with a 405 fluorophore | Biotium | 29028–1 | (1/50) |

## Zebrafish husbandry and embryo maintenance

Zebrafish husbandry used in all procedures was approved by the University of Kentucky Biosafety office as well as IACUC Policies, Procedures, and Guidelines and ethical standards. The AB strain was used as wildtype. ALT-R-CRISPR Cas9-based mutant lines were generated based on protocols established previously (*Piedade et al., 2020*). Embryos were kept at 28 °C in E3 embryo media. At indicated times in the study, embryos and adults were euthanized by cold shock or in tricaine before harvesting the eyes and fixation with 4% PFA in PBS overnight at 4 °C.

**Table 1.** WISH and genotyping primers used.

*WISH primers*

| | | |
|---|---|---|
| *cdhr1a* | ATGAAGAATGCAAGGGAAATA | TAATACGACTCACTATAGGGTCCTTCTGGACTGATTTCCAATGC |
| *pcdh15b* | GGTGATGGATCCAGTTCAGTG | TAATACGACTCACTATAGGGTCACAGAACAGTGGACTGAGA |
| *pcdh23* | AGTGATTCAGATGATCGACGA | TAATACGACTCACTATAGGGTCATAACTCTGTGATCTCTAA |
| *harmonin* | CCTTAGTTTGGTGGGCACCAA | TAATACGACTCACTATAGGGTTAAAAGAATGTCACCTCATC |
| *ush1ga* | TTCTTGCCTTAATGTCTGTTT | TAATACGACTCACTATAGGGGCGCAGCTTTCACAAAACCAT |
| *myo7aa* | AAACAAGGACATTTTAACCAC | TAATACGACTCACTATAGGGGAGTCACATCGATCACTGGAC |
| *GENOTYPING* | | |
| *cdhr1a* | GTGTTAAAATTTGAATGCTGAG | CTGCATATGCTTAGATGTTACC |
| *pcdh15b* | GAAACACAAAAGAAGCTGCG | GCCTTTATAATGGAGCGCAAG |

## Statistical analysis

All the data sets were analyzed using Prizm 10. Data shown on graphs represents individual measurements +/- standard deviation. Each of the data points had an n of 5+ individuals. One-way ANOVA with Tukey post hoc multiple comparisons analysis was used for assessing direct comparison significance between ages or genotypes. All of the graphs depict the mean value.

## Whole mount in situ hybridization

WISH was performed as previously described (*Holly et al., 2014*). RNA probes were generated via PCR amplification from 3 dpf cDNA fused to T7 promoter sequence and subsequently transcribed (DIG or FITC labeled) using T7 polymerase (Roche). Primer sequences can be found in *Table 1*. Approximately 1000 bp was amplified from the 3′ end of each cDNA. WISH was performed as previously described (*Holly et al., 2014*). Images were captured using a Nikon Digital sight DS-U3 camera and Elements software. Image adjustment was performed using Adobe Photoshop.

## Cryosection and immunofluorescence/IHC

Embryos and adult eyes were fixed in 4% paraformaldehyde then washed out PBS with 0.1%Tween-20. Next, the specimens were washed overnight in 10% then 30% sucrose overnight at 4 °C. Transverse, 10 mm sections were collected, beginning just anterior to and ending posterior to the eye. For imaging and cell quantification, only the sections containing the lens were used for consistency. Immunofluorescence was performed as described previously with addition of a Tris-EDTA-based antigen retrieval step prior to serum blocking the specimen (*Piedade et al., 2020*). Post blocking, following antibodies and lectins were used: anti-zcdhr1a (Cdhr1a, rabbit, 1:100, SKU:DZ07988 Boster Bio, Pleasonton, CA, United States), anti-hcdhr1 (CDHR1, rabbit, 1:100, PA5-57832 Thermo Fisher, Waltham, MA, United States), anti-hpcdh15 (Pcdh15, sheep, 1:75, PA5-47865 Thermo Fisher, Waltham, MA, United States), anti-prph2 (Peripherin – 2, rabbit, 1:100, 18109–1-AP Protein-tech, Rosemont, IL, United States), anti-actin (beta Actin, mouse, 1:100, MA1-140 Thermo Fisher, Waltham, MA, United States), anti-gnb1 (GNB1, rabbit, 1:100, PA5-30046 Thermo Fisher, Waltham, MA, United States), anti-zrhodopsin (1d1 epitope of the zebrafish rhodopsin, mouse 1:100, Fadool lab, Florida State University, FL, United States). Alexa Fluor 488, 555, and 647 conjugated secondary antibodies at a concentration of 1:200 (Thermo Fisher, Waltham, MA, United States) were used with DAPI as a counter stain. Further, Lectins: Peanut Germ Agglutinin conjugated with a 488 fluorophore (1:100 concentration, #29060 Biotium, San Francisco, CA, United States) and Wheat Germ Agglutinin conjugated with a 405 fluorophore (1:50 concentration, #29028–1 Biotium, San Francisco, CA, United States) were used to label PRC outer segments.

## Confocal/SIM imaging

For immunofluorescence, slides were mounted in Vectashield antifade mounting medium (H-1700–10 Vector laboratories, Newark, CA, United States) then imaged on a Nikon C2 confocal microscope under a 60X1.4 NA oil immersion objective. For consistency, all the images were captured from

the central retina to assess all the major types of photoreceptors. Super-resolution microscopy was performed on a Nikon A1R confocal microscope equipped with a Structured Illumination Microscopy module and a CMOS sensor under a 100X1.49 NA oil immersion objective. Quantification of OS length and number was performed manually using confocal captured images in Image J.

## Cloning

A previously developed and verified Cdhr1a mammalian cell expression plasmid of Cdhr1a tagged with FLAG in pCIG2 developed via Infusion HD cloning plus (Takara) was used in this study (*Piedade et al., 2020*). To develop an expression construct for Pcdh15b, we amplified pcdh15b cDNA from 3 dpf WT embryos (5'CATCATGATATCACCATGAAGATGCGCCAGAGGTCG-3', 5'ATGATGTTCGAA CAGAACAGTGGACTGAGATGG3') and cloned into the Topo XL vector (Invitrogen). Subsequently, we directionally cloned pcdh15b cDNA into the pCDNA3-MYC expression plasmid using Kpn1 and Xho1. Both constructs were verified using Sanger sequencing (Eurofins Genomics).

## Cell culture

We received the authenticated cells from ATCC (HEK293T: CRL-11268G-1 and K562: CLL-243). Cells were tested bi-weekly for mycoplasma using a PCR-based kit from Thermo Fisher (J66117.AMJ). Cells were grown in DMEM +10% FBS.

## Immunoprecipitation

For transient mammalian cell transfection, a previously established protocol was used with slight modifications (*Pereira Piedade et al., 2019*). First, HEK 293 cells were cultured at 37 °C in DMEM media until 70% confluency. Next, the aforementioned plasmids individually or in combinations of Cdhr1a:-FLAG pCIG2 (1 μg) and Pcdh15b:MYC pCDNA (1 μg) were first incubated with PEI (polyethyleneimine) at room temperature for 30 min. Upon HEK 293 cell confluency, these PEI/DNA complexes were transfected to the cells and incubated at 37 °C for 24 hr. Protein extraction was performed based on the Mem-PER plus extraction protocol (#89842 Thermo Fisher). To inhibit protease activity, we added HALT protease inhibitor (1:1000 #78425 Thermo Fisher). For co-immunoprecipitation (co-IP), one Cdhr1a and Pcdh15b cotransfected protein sample was incubated with Anti-FLAG magnetic beads (A36797 Thermo Fisher) while another co-transfected sample was incubated with Anti-MYC magnetic beads (#88842 Thermo Fisher). For IP control, Cdhr1a-FLAG transfection samples were immunoprecipitated (IP) using Anti-MYC beads while Pcdh15b-MYC transfection samples were immunoprecipitated using Anti-FLAG beads. Next, all the lysate control, control IP, and co-IP samples were prepared with 6 X SDS sample buffer under reducing conditions and boiled at 96 °C for 7 min. Samples were then run in two 7.5% SDS PAGE gels at 140 volts for 90 min. They were then transferred to a PVDF membrane for 90 min at 20 volts. The membranes were blocked with 1 X Blocker (#37565 Thermo Fisher) for 1 hr at room temperature. Next, one membrane was probed for FLAG via mouse Anti-FLAG (1:1000 F1804 Sigma Aldrich, St. Louis, MO, United States) and the other membrane was probed for MYC via mouse Anti-MYC (1:1000 MA121316 Thermo Fisher). These primary antibodies were incubated after blocking at 4 °C overnight. The following day, membranes were washed three times with TBS (5 min each). Subsequently, the membranes were incubated with goat anti-mouse secondary antibody conjugated with poly HRP at 1:2000 concentration in the 1 X Blocker for 1 hr at room temperature. Next, the membranes were again washed with TBS buffer three times (5 min each). To detect the HRP signal, SuperSignal West Pico PLUS Chemiluminescent Substrate (#34580 Thermo Fisher) was used and chemiluminescence images were then detected and captured using Amersham Imager 680.

## K562 assay

For the cell aggregation assay, K562 cells were cultured in plastic dishes at 37 °C in RPMI 1640 media until 70% confluency. Cdhr1a and Pcdh15b expression plasmids were incubated with the PLUS reagent for 15 min in the same RPMI 1640 media and then Lipofectamine LTX was added and incubated for 25 min at room temperature to form Lipofectamine-DNA complexes. The complexes were then transferred to the cells and incubated at 37 °C for 24 hr. The next day, individual transfections of Cdhr1a, Pcdh15b, or control were co-cultured by isolating each population of cells from the dishes, mixing them together, and re-plating to allow aggregates to form. Any K562 cell aggregates that formed were then imaged 24 hr later using the EVOS FLoid cell culture microscope (#4471136

Thermo Fisher). The experiment was repeated three times and images from five different fields of view of each replicate were quantified.

## TEM

TEM sample preparation was slightly modified from *Miles et al., 2021*. First, zebrafish larvae and adult specimens were fixed in 2.5% glutaraldehyde and 2% paraformaldehyde in 0.1 M phosphate buffer at 4 °C overnight on a shaker. Samples were then washed five times with 0.1 M phosphate buffer (with 0.1% Tween-20) for five times - 10 min each to wash out the aldehyde fixatives. Next, samples were placed in 1% Osmium tetroxide solution for 1.5 hr in dark at room temperature on a shaker. Samples were then washed again with 0.1 M phosphate buffer with Tween-20 for five times – 5 min each and subsequently dehydrated with a series of ethanol washes at 50% (10 min), 70% (10 min), 80% (10 min), 90% (15 min), and two 100% ethanol washes (15 min each). Next, the specimens were infiltrated with LR white resin through a series of ethanol:resin combinations 25% LR white: 75% ethanol (30 min), 50% LR white: 50% ethanol (30 min), 75% LR white: 25% ethanol (1 hr), and with 100% LR white (overnight at 4 °C). The following day, samples were exchanged with fresh 100% LR white for 3 hr and then samples were placed in '00' gelatin capsules or sealed PCR tubes in order to perform anaerobic polymerization required for LR white at 65 °C. One-micron-thin sections were collected via a glass blade from the resin blocks using Leica EM UC7 and were stained with Toluidine blue to assess the location of samples being sectioned. Once satisfied with the location, ultrathin 70 nm sections were collected using a diamond knife (Ultra 45° DiATOME, Quakertown, PA, United States) and placed on slotted EM grids (G2010-Ni Electron Microscopy Sciences, Hatfield, PA, United States) covered in a layer of 0.5% formvar. Sections were then stained with a non-radioactive heavy metal UranyLess EM stain (#22409 Electron Microscopy Sciences) for 1 hr followed by Reynold's lead citrate stain for 30 min. Images were then collected using FEI Talos F200X at 200 keV.

## Immuno-TEM

5 dpf embryos were fixed in 4% paraformaldehyde then washed out using PBS with 0.1% Tween-20. Next, eyes were enucleated from embryos and were processed for whole-mount immunohisto-chemistry (IHC). Whole-mount IHC was performed as described previously, but with the exception where heat-mediated antigen retrieval for whole eyes was conducted in a thermal cycler prior to serum blocking. Post-blocking, primary antibodies used were: anti-zcdhr1a (SKU:DZ07988 Boster Bio, Pleasonton, CA, United States), and anti-hpcdh15 (PA5-47865 Thermo Fisher, Waltham, MA, United States). Gold-conjugated secondary antibodies used were: goat anti-Rabbit Alexa Fluor 488–5 nm colloidal gold (A-31565, Thermo Fisher, Waltham, MA, United States) and Donkey-Anti-Sheep 25 nm colloidal gold (SKU: 25829 Electron Microscopy Sciences, Hatfield, PA, United States). Gold-labeled eyes were then processed for TEM as described previously.

## CRISPR

To establish the CRISPR/Cas9-based mutant lines, we ordered predesigned and synthesized ALT-R Crispr crRNA (IDTDNA) targeting Cdhr1a (GTCTGGAAGTAGCATCTATA, TCTGGCACATCTACGATGGA) and Pcdh15b (CACCACAATGGACTGGATGT, CGACTATCCGCACCTCGTGT). Next, crRNA-tracrRNA duplexes were constructed before injecting with Alt-R Cas9 v.3 enzyme in single-cell staged embryos as described previously (*Hoshijima et al., 2019*). Genotyping was performed by designing primer oligos upstream and downstream of the expected CRISPR/Cas9-based deletion sites and the resulting difference in sequence size was used to genotype using DNA gel electrophoresis. (Cdhr1a: 5'-GTGTTAAAATTTGAATGCTGAG-3', 5'-CTGCATATGCTTAGATGTTACC-3',pcdh15b: 5'-GAAA CACAAAAGAAGCTGCG-3', 5'- GCCTTTATAATGGAGCGCAAG-3') Both Cdhr1a and Pcdh15b dele-tion sequences were then confirmed via Sanger sequencing after outcrossing for two filial generations.

## Acknowledgements

We thank Dr. Ann Morris for insightful comments and review of the manuscript. Thanks to Dr. Chintan Kikani for providing tissue culture facilities. TEM work was performed in part at the U.K. Electron microscopy core, a member of the National Nanotechnology Coordinated Infrastructure (NNCI), which is supported by the National Science Foundation (NNCI-2025075), and we thank Jillian Cramer and Dr. Nicolas Briot for technical assistance. SIM was performed at the UKY Light microscopy core,

and we thank Dr. Xu Fu for technical assistance. This work was supported by a grant from the Retina Research Foundation awarded to JKF, KYINBRE core facility voucher (NIH Award P20GM103436) and the Gertrude Ribble Pilot Grant (department of Biology, University of Kentucky) awarded to MP.

## Additional information

### Funding

| Funder | Grant reference number | Author |
| --- | --- | --- |
| Retina Research Foundation | | Jakub K Famulski |
| KYINBRE core facility voucher | P20GM103436 | Jakub K Famulski |
| Gertrude Ribble Pilot Grant | | Meet K Patel |

The funders had no role in study design, data collection and interpretation, or the decision to submit the work for publication.

### Author contributions

Meet K Patel, Conceptualization, Data curation, Formal analysis, Investigation, Methodology, Writing – original draft, Writing – review and editing; Warlen Pereira Piedade, Methodology, Writing – review and editing; Jakub K Famulski, Conceptualization, Supervision, Funding acquisition, Methodology, Writing – original draft, Project administration, Writing – review and editing

### Author ORCIDs

Jakub K Famulski ⓘ https://orcid.org/0000-0002-3619-4113

### Ethics

Zebrafish husbandry and experimentation used in all procedures were approved by the University of Kentucky Biosafety office as well as IACUC Policies, Procedures, and Guidelines and ethical standards. IACUC protocol 2021-3781.

Reviewer #1 (Public review): https://doi.org/10.7554/eLife.102258.3.sa1
Reviewer #2 (Public review): https://doi.org/10.7554/eLife.102258.3.sa2
Reviewer #3 (Public review): https://doi.org/10.7554/eLife.102258.3.sa3
Author response https://doi.org/10.7554/eLife.102258.3.sa4

## Additional files

### Supplementary files

MDAR checklist

Source data 1. This source data contains all the individual counts and measurements included in the manuscript.

### Data availability

All data generated or analyzed during this study are included in this published article, source data files have been provided.

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
