## [Editor Report · eLife Assessment]

This **valuable** study investigates the interaction of two integral membrane proteins (Cdhr1a and Pcdh15b) and their roles in cone-rod dystrophy. **Convincing** evidence using loss-of-function mutants demonstrates clearly that both proteins are required for cone maintenance and survival. Although some evidence (Western blots and cell aggregation assays) demonstrates Cdhr1a and Pcdh15b can physically interact, there is insufficient evidence to support the subcellular localization and the proposed heterodimeric interaction of the two proteins from distinct subcellular compartments in cone photoreceptors.

---

## [Referee Report · Reviewer #1 (Public review)]

Mutations in CDHR1, the human gene encoding an atypical cadherin-related protein expressed in photoreceptors, are thought to cause cone-rod dystrophy (CRD). However, the pathogenesis leading do this disease is unknown. Previous work has led to the hypothesis that CDHR1 is part of a cadherin-based junction that facilitates the development of new membranous discs at the base of the photoreceptor outer segments, without which photoreceptors malfunction and ultimately degenerate. CDHR1 is hypothesized to bind to a transmembrane partner to accomplish this function, but the putative partner protein has yet to be identified.

The manuscript by Patel et al. makes an important contribution toward improving our understanding of the cellular and molecular basis of CDHR1 associated CRD. Using gene editing, they generate a loss of function mutation in the zebrafish cdhr1a gene, an ortholog of human CDHR1, and show that this novel mutant model has a retinal dystrophy phenotype, specifically related to defective growth and organization of photoreceptor outer segments (OS) and calyceal processes (CP). This phenotype seems to be progressive with age. Importantly, Patel et al, present intriguing evidence that pcdh15b, also known for causing retinal dystrophy in previous Xenopus and zebrafish loss of function studies, is the putative cdhr1a partner protein mediating the function of the junctional complex that regulates photoreceptor OS growth and stability.

This research is significant in that it:

(1) provides evidence for a progressive, dystrophic photoreceptor phenotype in the cdhr1a mutant and, therefore, effectively models human CRD; and

(2) identifies pcdh15b as the putative, and long sought after, binding partner for cdhr1a, further supporting the theory of a cadherin-based junction complex that facilitates OS disc biogenesis.

Comments on the revised version of the manuscript:

The authors adequately addressed previous comments related to lack of details on quantitative and statistical analyses and methods. In this regard, I believe the revised manuscript presents a stronger analysis of the data. I also appreciated the revised discussion section, which better contextualizes their new data with previous observations in different animal models.

The authors provided additional evidence in Fig 1C-H for the co-localization of pcdh15b and actin within CPs using immunolabeling with super resolution imaging. This data firmly supports their other observations. A similar approach tends to also show co-localization of actin and cdhr1a, although the authors suggest that the pattern of expression is less overlapping, which would be expected if cdhr1a is predominately expressed in the OS membranes whereas pcdh15b is predominantly expressed in the CP membranes. In my opinion the data presented to support this separation is still not that convincing. Moreover, the authors show that both cdhr1a and pcdh15b are expressed in CPs using immuno-TEM (Fig 1I). This is a difficult question to address experimentally, and it is, of course, still plausible that pcdh15b within the CP membrane and cdhr1a within the OS membrane are interacting in trans. However, I just don't think that the data unequivocally support mutually exclusive localization of these proteins as suggested by the authors and depicted in the model in Fig 1J.

---

## [Referee Report · Reviewer #2 (Public review)]

Summary:

The goal of this study was to develop a model for CDHR1-based Con-rod dystrophy and study the role of this cadherin in cone photoreceptors. Using genetic manipulation, a cell binging assay, and high- resolution microscopy the authors find that like rods, cones localize CDHR1 to the lateral edge of outer segment (OS) discs and closely opposes PCDH15b which is known to localize to calyceal processes (CPs). Ectopic expression of CDHR1 and PCDH15b in K652 cells indicate these cadherins promote cell aggregation as heterophilic interactants, but not through homophilic binding. This data suggests a model where CDHR1 and PCDH15b link OS and CPs and potential stabilize cone photoreceptor structure. Mutation analysis of each cadherin results in cone structural defects at late larval stages. While pcdh15b homozygous mutants are lethal, cdhr1 mutants are viable and subsequently show photoreceptor degeneration by 3-6 months.

Strengths:

A major strength of this research is the development of an animal model to study the cone specific phenotypes associate with CDHR1-based CRD. The data supporting CDHR1 (OS) and PCDH15 (CP) binding is also a strength, although this interaction could be better characterized in future studies. The quality of the high-resolution imaging (at the light and EM levels) is outstanding. In general, results support the conclusions of the authors.

Weaknesses:

While the cellular phenotyping is strong, the functional consequences of CDHR1 disruption is not addressed. While this is not the focus of the investigation, such analysis would raise the impact of the study overall. This is particularly important given some of the small changes observed in OS and CP structure. While statistically significant, are the subtle changes biologically significant? Examples include cone OS length (Fig 4F, 6E) as well as other morphometric data (Fig 7I in particular). Related, for quantitative data and analysis throughout the manuscript, more information regarding the number of fish/eyes analyzed as well as cells per sample would provide confidence in the rigor. The authors should also not whether analysis was done in an automated and/or masked manner.

Comments on revisions:

Most of my concerns were addressed in this revised version.

---

## [Referee Report · Reviewer #3 (Public review)]

Summary:

The manuscript by Patel et al investigates the hypothesis that CDHR1a on photoreceptor outer segments is the binding partner for PCDH15 on the calyceal processes, and the absence of either adhesion molecule results in separation between the two structures, eventually leading to degeneration. PCDH15 mutations cause Usher syndrome, a disease of combined hearing and vision loss. In the ear, PCDH15 binds CDH23 to form tip links between stereocilia. The vision loss in less understood. Previous work suggested PCDH15 is localized to the calyceal processes, but the expression of CDH23 is inconsistent between species. Patel et al suggest that CDHR1a (formerly PCDH21) fulfills the role of CDH23 in the retina.

The experiments are mainly performed using the zebrafish model system. Expression of Pcdh15b and Cdhr1a protein is shown in the photoreceptor layer through standard confocal and structured illumination microscopy. The two proteins co-IP and can induce aggregation in vitro. Loss of either Cdhr1a or Pcdh15, or both, results in degeneration of photoreceptor outer segments over time, with cones affected primarily.

The idea of the study is logical given the photoreceptor diseases caused by mutations in either gene, the comparisons to stereocilia tip links, and the protein localization near the outer segments. The work here demonstrates that the two proteins interact in vitro and are both required for ongoing outer segment maintenance. The major novelty for this paper would be the demonstration that Pcdh15 localized to calyceal processes interacts with Cdhr1a on the outer segment, thereby connecting the two structures. Unfortunately, the data as presented are inadequate proof of this model.

Strengths:

The in vitro data to support the ability of of Pcdh15b and Cdhr1a to bind is well done. The use of pcdh15b and cdhr1a single and double mutants is also a strength of the study, especially being that this would be the first characterization of a zebrafish cdhr1a mutant.

This is a large body of data.

Weaknesses:

(1) I have serious concerns about the quality of the imaging here. The premise that cdhr1a/pcdh15 juxtaposition is evidence for the two proteins mediating the connection between outer segments and calyceal processes requires very careful microscopy. The SIM images have two major issues - one being that the red and green channels are misaligned and the other being evidence of bleed through between the channels. This is obvious in Fig 2A but likely true across all the panels in Fig 2, and possibly applies to confocal images in Fig 1 as well. The co-labelling with actin shows very uneven, punctate staining for actin bundles.

(2) The newly added TEM and transverse sections include colored regions that obscure the imaging.

(3) The quantification should be done with averages from individual fish. Counting individual measurements as single data points artificially inflates the significance. Also, the cone subtypes are still lumped together for analysis despite their variable sizes.

(4) I highlighted previously that the measurement of calyceal processes was incorrect. The redrawn labels in Fig 7 are now more accurate, although still difficult to interpret. However, the quantification in Fig 7O is exactly the same. How can that be if the measurement region is now different?

(5) Lower magnification views would provide context for the TEM data.

(6) The statement describing the separation between calyceal processes and the outer segment in the mutants is still not backed up by the data.

(7) The authors state "from the fact that rod CPs are inherently much smaller than cone CPs". This is now referenced, but incorrectly. Also, the issue of pigment interference was not addressed.

(8) The images in panels B-F of the Supplemental Figure are uncannily similar, possibly even of the same fish at different focal planes.

---

## [Author Response]

The following is the authors’ response to the original reviews.

**Public Reviews:**

**Reviewer #1 (Public review):**
Mutations in CDHR1, the human gene encoding an atypical cadherin-related protein expressed in photoreceptors, are thought to cause cone-rod dystrophy (CRD). However, the pathogenesis leading to this disease is unknown. Previous work has led to the hypothesis that CDHR1 is part of a cadherin-based junction that facilitates the development of new membranous discs at the base of the photoreceptor outer segments, without which photoreceptors malfunction and ultimately degenerate. CDHR1 is hypothesized to bind to a transmembrane partner to accomplish this function, but the putative partner protein has yet to be identified.The manuscript by Patel et al.makes an important contribution toward improving our understanding of the cellular and molecular basis of CDHR1-associated CRD. Using gene editing, they generate a loss of function mutation in the zebrafish cdhr1a gene, an ortholog of human CDHR1, and show that this novel mutant model has a retinal dystrophy phenotype, specifically related to defective growth and organization of photoreceptor outer segments (OS) and calyceal processes (CP). This phenotype seems to be progressive with age. Importantly, Patel et al, present intriguing evidence that pcdh15b, also known for causing retinal dystrophy in previous Xenopus and zebrafish loss of function studies, is the putative cdhr1a partner protein mediating the function of the junctional complex that regulates photoreceptor OS growth and stability.This research is significant in that it:(1) Provides evidence for a progressive, dystrophic photoreceptor phenotype in the cdhr1a mutant and, therefore, effectively models human CRD; and(2) Identifies pcdh15b as the putative, and long sought after, binding partner for cdhr1a, further supporting the theory of a cadherin-based junction complex that facilitates OS disc biogenesis.Nonetheless, the study has several shortcomings in methodology, analysis, and conceptual insight, which limits its overall impact.Below I outline several issues that the authors should address to strengthen their findings.Major comments:(1) Co-localization of cdhr1a and pcdh15b proteinsThe model proposed by the authors is that the interaction of cdhr1a and pcdh15b occurs in trans as a heterodimer. In cochlear hair cells, PCDH15 and CDHR23 are proposed to interact first as dimers in cis and then as heteromeric complexes in trans. This was not shown here for cdhr1a and pcdh15b, but it is a plausible configuration, as are single heteromeric dimers or homodimers. Regardless, this model depends on the differential compartmental expression of the cdhr1a and pcdh15b proteins. Data in Figure 1 show convincing evidence that these two proteins can, at least in some cases, be distributed along the length of photoreceptor membranes that are juxtaposed, as would be the case for OS and CP. If pcdh15b is predominantly expressed in CPs, whereas cdhr1a is predominantly expressed in OS, then this should be confirmed with actin double labeling with cdhr1a and pcdh15b since the apicobasal oriented (vertical) CPs would express actin in this same orientation but not in the OS. This would help to clarify whether cdhr1a and pcdh15b can be trafficked to both OS and CP compartments or whether they are mutually exclusive.

First let me thank the reviewer for taking the time to comprehensively evaluate our work and provide constructive criticism which will improve the quality of our final version.

To address this issue, we are completed imaging of actin/cdhr1a and actin/pcdh15b using SIM in both transverse and axial sections (Fig 1C-H). Additionally, we have recently established an immuno-gold-TEM protocol and showcase co-labeling of cdhr1a and pcdh15b at TEM resolution along the CP (Fig 1I).

Photoreceptor heterogeneity goes beyond the cone versus rod subtypes discussed here and it is known that in zebrafish, CP morphology is distinct in different cone subtypes as well as cone versus rod. It would be important to know which specific photoreceptor subtypes are shown in zebrafish (Figures 1A-C) and the non-fish species depicted in Figures 1E-L. Also, a larger field of view of the staining patterns for Figures 1E-L would be a helpful comparison (could be added as a supplementary figure).

The revised manuscript includes labels for the location of different cone subtypes in figure 1. All of the images showcasing CHDR1 localization across species concentrate on the PNA positive R/G cones. Larger fields of view were not collected as we prioritized the highest resolution possible and therefore collected small fields of view.

(2) Cdhr1a function in cell cultureThe authors should explain the multiple bands in the anti-FLAG blots. Also, it would be interesting to confirm that the cdhr1a D173 mutant prevents the IP interaction with pcdh15b as well as the additive effects in aggregate assays of Figure 2.

The multiple bands on the WB is like our previous results (Piedade 2020), which we believe arise due to ubiquitination and proteolytic cleavage of cdhr1a. We expect the D173 mutation to result in a complete absence of cdhr1a polypeptide, based on the lack of in situ signal in our WISH studies.

Is it possible that the cultured cells undergo proliferation in the aggregation assays shown in Figure 2? Cells might differentially proliferate as clusters form in rotating cultures. A simple assay for cell proliferation under the different transfection conditions showing no differences would address this issue and lend further support to the proposed specific changes to cell adhesion as a readout of this assay.

This is a possibility; however we did not use rotating cultures, this was a monolayer culture. We did not observe any differences in total cell number between the differing transfections. As such, we do not feel proliferation explains the aggregation of K562 cells.

Also, the authors report that the number of clusters was normalized to the field of view, but this was not defined. Were the n values different fields of view from one transfection experiment, or were they different fields of view from separate transfection experiments? More details and clarification are needed.

This will be clarified in the revised manuscript, in short we replicated this experiment 3 times, quantifying 5 different fields of view in each replicate.

(3) Methodological issues in quantification and statistical analysesWere all the OS and CP lengths counted in the observation region or just a sample within the region? If the latter, what were the sampling criteria? For CPs, it seems that the length was an average estimate based on all CPs observed surrounding one cone or one-rod cell. Is this correct? Again, if sampled, how was this implemented? In Fig 4M', the cdhr1a-/- ROS mostly looks curvilinear. Did the measurements account for this, or were they straight linear dimension measurements from base to tip of the OS as depicted in Fig 5A-E? A clearer explanation of the OS and CP length quantification methodology is required.

The revised manuscript will clearly outline measurement methods. In short, we measured every CP/OS in the imaged regions. We did not average CPs/cell, we simply included all CP measurements in our analysis. All our CP measurements (actin or cdhr1a or pcdh15), were measured in the presence of a counter stain, WGA, prph2, gnb1 or PNA to ensure proper measurements (landmark) and association with proper cell type. Our new figure 7 now includes cone OS counter staining to better highlight the OS.

All measurements were taken as best as possible to reflect a straight linear dimension for consistency.

How were cone and rod photoreceptor cell counts performed? The legend in Figure 4 states that they again counted cells in the observation region, but no details were provided. For example, were cones and rods counted as an absolute number of cells in the observation region (e.g., number of cones per defined area) or relative to total (DAPI+) cell nuclei in the region? Changes in cell density in the mutant (smaller eye or thinner ONL) might affect this quantification so it would be important to know how cell quantification was normalized.

The revised manuscript will clearly outline measurement methods. In short, rod and cone cell counts were based on the number of outer segments that were observed in the imaging region and previously measured for length. We did not observe any eye size differences in our mutant fish.

In Figure 6I, K, measuring the length of the signal seems problematic. The dimension of staining is not always in the apicobasal (vertical) orientation. It might be more accurate to measure the cdhr1a expression domain relative to the OS (since the length of the OS is already reduced in the mutants). Another possible approach could be to measure the intensity of cdhr1 staining relative to the intensity within a Prph2 expression domain in each group. The authors should provide complementary evidence to support their conclusion.

The revised manuscript will clearly outline measurement methods. In short, all of our CP measurements (actin or cdhr1a or pcdh15), were done in the presence of a counter stain, WGA, prph2, gnb1 or PNA to ensure proper measurements and association with proper cell type.

A better description of the statistical methodology is required. For example, the authors state that "each of the data points has an n of 5+ individuals." This is confusing and could indicate that in Figure 4F alone there were ~5000 individuals assayed (~100 data points per treatment group x n=5 individuals per data point x 10 treatment groups). I don't think that is what the authors intended. It would be clearer if the authors stated how many OS, CP, or cells were counted in their observation region averaged per individual and then provided the n value of individuals used per treatment group (controls and mutants), on which the statistical analyses should be based.

This has been addressed in the revised manuscript. In short, we had an n=5 (individual fish) analyzed for each genotype/time point.

There are hundreds of data points in the separate treatment groups shown in several of the graphs. It would not be correct to perform the ANOVA on the separate OS or CP length measurements alone as this will bias the estimates since they are not all independent samples. For example, in Figure 6H, 5dpf pcdh15b+/- have shorter CPs compared to WT but pcdh15b-/- have longer compared to WT. This could be an artifact of the analysis. Moreover, the authors should clarify in the Methods section which ANOVA post hoc tests were used to control for multiple pairwise comparisons.

We have re-analyzed the data using multiple pairwise comparison ANOVA with post hoc tests (Tukey test). This new analysis did not significantly alter the statistical significance outcome of the study.

(4) Cdhr1a function in photoreceptorsThe Cdhr1a IHC staining in 5dpf WT larvae in Figure 3E appears different from the cdhr1a IHC staining in 5dpf WT larvae in Figure 1A or Figure 6I. Perhaps this is just the choice of image. Can the authors comment or provide a more representative image?

The image in figure 3E was captured using a previous non antigen retrieval protocol which limits the resolution of the cdhr1a signal along the CP. In the revised manuscript we have included an image that better represents cdhr1a staining in the WT and mutant.

The authors show that pcdh15b localization after 5dpf mirrored the disorganization of the CP observed with actin staining. They also show in Figure 5O that at 180dpf, very little pcdh15b signal remains. They suggest based on this data that total degradation of CPs has occurred in the cdhr1a-/- photoreceptors by this time. However, although reduced in length, COS and cone CPs are still present at 180dpf (Figure 5E, E'). Thus, contrary to the authors' general conclusion, it is possible that the localization, trafficking, and/or turnover of pcdh15b is maintained through a cdhr1a-dependent mechanism, irrespective of the degree to which CPs are maintained. The experiments presented here do not clearly distinguish between a requirement for maintenance of localization versus a secondary loss of localization due to defective CPs.

We agree, this point has been addressed in our revised manuscript. Additionally, we have also included data from 1 and 2 year old samples.

(5) Conceptual insightsThe authors claim that cdhr1a and pcdh15b double mutants have synergistic OS and CP phenotypes. I think this interpretation should be revisited.First, assuming the model of cdhr1a-pcdh15b interaction in trans is correct, the authors have not adequately explained the logic of why disrupting one side of this interaction in a single mutant would not give the same severity of phenotype as disrupting both sides of this interaction in a double mutant.Second, and perhaps more critically, at 10dpf the OS and CP lengths in cdhr1a-/- mutants (Figure 7J, T) are significantly increased compared to WT. In contrast, there are no significant differences in these measurements in the pcdh15b-/- mutants. Yet in double homozygous mutants, there is a significant reduction of ~50% in these measurements compared to WT. A synergistic phenotype would imply that each mutant causes a change in the same direction and that the magnitude of this change is beyond additive in the double mutants (but still in the same direction). Instead, I would argue that the data presented in Figure 7 suggest that there might be a functionally antagonistic interaction between cdhr1a and pcdh15b with respect to OS and CP growth at 10dpf.If these proteins physically interacted in vivo, it would appear that the interaction is complex and that this interaction underlies both OS growth-promoting and growth-restraining (stabilizing) mechanisms working in concert. Perhaps separate homodimers or heterodimers subserve distinct CP-OS functional interactions. This might explain the age-dependent differences in mutant CP and OS length phenotypes if these mechanisms are temporally dynamic or exhibit distinct OS growth versus maintenance phases. Regardless of my speculations, the model presented by the authors appears to be too simplistic to explain the data.

We agree with the reviewer, as such we have revised the discussion in our revised manuscript.

**Reviewer #2 (Public review):**
Summary:The goal of this study was to develop a model for CDHR1-based Con-rod dystrophy and study the role of this cadherin in cone photoreceptors. Using genetic manipulation, a cell binding assay, and high-resolution microscopy the authors find that like rods, cones localize CDHR1 to the lateral edge of outer segment (OS) discs and closely oppose PCDH15b which is known to localize to calyceal processes (CPs). Ectopic expression of CDHR1 and PCDH15b in K652 cells indicates these cadherins promote cell aggregation as heterophilic interactants, but not through homophilic binding. This data suggests a model where CDHR1 and PCDH15b link OS and CPs and potentially stabilize cone photoreceptor structure. Mutation analysis of each cadherin results in cone structural defects at late larval stages. While pcdh15b homozygous mutants are lethal, cdhr1 mutants are viable and subsequently show photoreceptor degeneration by 3-6 months.Strengths:A major strength of this research is the development of an animal model to study the cone-specific phenotypes associated with CDHR1-based CRD. The data supporting CDHR1 (OS) and PCDH15 (CP) binding is also a strength, although this interaction could be better characterized in future studies. The quality of the high-resolution imaging (at the light and EM levels) is outstanding. In general, the results support the conclusions of the authors.Weaknesses:While the cellular phenotyping is strong, the functional consequences of CDHR1 disruption are not addressed. While this is not the focus of the investigation, such analysis would raise the impact of the study overall. This is particularly important given some of the small changes observed in OS and CP structure. While statistically significant, are the subtle changes biologically significant? Examples include cone OS length (Figures 4F, 6E) as well as other morphometric data (Figure 7I in particular). Related, for quantitative data and analysis throughout the manuscript, more information regarding the number of fish/eyes analyzed as well as cells per sample would provide confidence in the rigor. The authors should also note whether the analysis was done in an automated and/or masked manner.

First let me thank the reviewer for taking the time to comprehensively evaluate our work and provide constructive criticism which will improve the quality of our final version.

The revised manuscript outlines both methods and statistics used for quantitation of our data. (please see comments from reviewer 1). While we do not include direct evidence of the mechanism of CDHR1 function, we do propose that its role is important in anchoring the CP and the OS, particularly in the cones, while in rods it may serve to regulate the release of newly formed disks (as previously proposed in mice). We do plan to test both of these hypothesis directly, however, that will be the basis of our future studies.

**Reviewer #3 (Public review):**
Summary:The manuscript by Patel et al investigates the hypothesis that CDHR1a on photoreceptor outer segments is the binding partner for PCDH15 on the calyceal processes, and the absence of either adhesion molecule results in separation between the two structures, eventually leading to degeneration. PCDH15 mutations cause Usher syndrome, a disease of combined hearing and vision loss. In the ear, PCDH15 binds CDH23 to form tip links between stereocilia. The vision loss is less understood. Previous work suggested PCDH15 is localized to the calyceal processes, but the expression of CDH23 is inconsistent between species. Patel et al suggest that CDHR1a (formerly PCDH21) fulfills the role of CDH23 in the retina.The experiments are mainly performed using the zebrafish model system. Expression of Pcdh15b and Cdhr1a protein is shown in the photoreceptor layer through standard confocal and structured illumination microscopy. The two proteins co-IP and can induce aggregation in vitro. Loss of either Cdhr1a or Pcdh15, or both, results in degeneration of photoreceptor outer segments over time, with cones affected primarily.The idea of the study is logical given the photoreceptor diseases caused by mutations in either gene, the comparisons to stereocilia tip links, and the protein localization near the outer segments. The work here demonstrates that the two proteins interact in vitro and are both required for ongoing outer segment maintenance. The major novelty of this paper would be the demonstration that Pcdh15 localized to calyceal processes interacts with Cdhr1a on the outer segment, thereby connecting the two structures. Unfortunately, the data presented are inadequate proof of this model.Strengths:The in vitro data to support the ability of Pcdh15b and Cdhr1a to bind is well done. The use of pcdh15b and cdhr1a single and double mutants is also a strength of the study, especially being that this would be the first characterization of a zebrafish cdhr1a mutant.Weaknesses:(1) The imaging data in Figure 1 is insufficient to show the specific localization of Pcdh15 to calyceal processes or Cdhr1a to the outer segment membrane. The addition of actin co-labelling with Pcdh15/Cdhr1a would be a good start, as would axial sections. The division into rod and cone-specific imaging panels is confusing because the two cell types are in close physical proximity at 5 dpf, but the cone Cdhr1a expression is somehow missing in the rod images. The SIM data appear to be disrupted by chromatic aberration but also have no context. In the zebrafish image, the lines of Pcdh15/Cdhr1a expression would be 40-50 um in length if the scale bar is correct, which is much longer than the outer segments at this stage and therefore hard to explain.

First let me thank the reviewer for taking the time to comprehensively evaluate our work and provide constructive criticism which will improve the quality of our final version.

To address this issue, we have added images of actin/cdhr1a and actin/pcdh15b using SIM in both transverse and axial sections. Additionally, we have established an immuno-gold-TEM protocol and provide data showcasing co-labeling of cdhr1a and pcdh15b at TEM resolution.

(2) Figure 3E staining of Cdhr1a looks very different from the staining in Figure 1. It is unclear what the authors are proposing as to the localization of Cdhr1a. In the lab's previous paper, they describe Cdhr1a as being associated with the connecting cilium and nascent OS discs, and fail to address how that reconciles with the new model of mediating CP-OS interaction. And whether Cdhr1a localizes to discrete domains on the disc edges, where it interacts with Pcdh15 on individual calyceal processes.

The image in figure 3E was captured using a previous non antigen retrieval protocol which limits the resolution of the cdhr1a signal along the CP. In the revised manuscript we include an image that better represents cdhr1a staining in the WT and mutant.

(3) The authors state "In PRCs, Pcdh15 has been unequivocally shown to be localized in the CPs". However, the immunostaining here does not match the pattern seen in the Miles et al 2021 paper, which used a different antibody. Both showed loss of staining in pcdh15b mutants so unclear how to reconcile the two patterns.

We agree that our staining appears different, but we attribute this to our antigen retrieval protocol which differed from the Miles et al paper. We also point to the fact that pcdh15b localization has been shown to be similar to our images in other species (monkey and frog). As such, we believe our protocol reveals the proper localization pattern which might be lost/hampered in the procedure used in Miles et al 2021.

(4) The explanation for the CRISPR targets for cdhr1a and the diagram in Figure 3 does not fit with crRNA sequences or the mutation as shown. The mutation spans from the latter part of exon 5 to the initial portion of exon 6, removing intron 5-6. It should nevertheless be a frameshift mutation but requires proper documentation.

This was an overlooked error in figure making, we have corrected this typo in the revised manuscript.

(5) There are complications with the quantification of data. First, the number of fish analyzed for each experiment is not provided, nor is the justification for performing statistics on individual cell measurements rather than using averages for individual fish. Second, all cone subtypes are lumped together for analysis despite their variable sizes. Third, t-tests are inappropriately used for post-hoc analysis of ANOVA calculations.

As we discussed for reviewer 1 and 2, all methods and quantification/statistics will be clearly described in the revised manuscript.

(6) Unclear how calyceal process length is being measured. The cone measurements are shown as starting at the external limiting membrane, which is not equivalent to the origin of calyceal processes, and it is uncertain what defines the apical limit given the multiple subtypes of cones. In Figure 5, the lines demonstrating the measurements seem inconsistently placed.

As we discussed for reviewer 1 and 2, all methods and quantification/statistics will be clearly described in the revised manuscript. We have also clarified that CP measurements were made based on a counterstain for the cone/rod OS so that the actin signal was only CP associated. We have included the counter stain in our revised Figure 7.

(7) The number of fish analyzed by TEM and the prevalence of the phenotype across cells are not provided. A lower magnification view would provide context. Also, the authors should explain whether or not overgrowth of basal discs was observed, as seen previously in cdhr1-null frogs (Carr et al., 2021).

The revised manuscript now includes the n number for our TEM samples. We have also added text comparing our results directly to Carr 2021.

(8) The statement describing the separation between calyceal processes and the outer segment in the mutants is not backed up by the data. TEM or co-labelling of the structures in SIM could be done to provide evidence.

We have completed both more SIM as well as immuno-gold TEM to support our conclusions, see new Figure 1.

(9) "Based on work in the murine model and our own observations of rod CPs, we hypothesize that zebrafish rod CPs only extend along the newly forming OS discs and do not provide structural support to the ROS." Unclear how murine work would support that conclusion given the lack of CPs in mice, or what data in the manuscript supports this conclusion.

In the revised manuscript we have adjusted our discussion to hypothesize that the small length of rod CPs is most likely to represent their interaction with newly forming discs rather than connect with mature discs which are enclosed in the OS.

(10) The authors state "from the fact that rod CPs are inherently much smaller than cone CPs" without providing a reference. In the manuscript, the measurements do show rod CPs to be shorter, but there are errors in the cone measurements, and it is possible that the RPE pigment is interfering with the rod measurements.

We have included references where rod CPs have been found to be shorter. We have no doubt that in zebrafish the rod CPs are significantly shorter. All our CP measurements are done with a counter stain for rods and cones to be sure that we are measuring the correct cell type.

(11) The discussion should include a better comparison of the results with ocular phenotypes in previously generated pcdh15 and cdhr1 mutant animals.

The revised manuscript has included these points.

(12) The images in panels B-F of the Supplemental Figure are uncannily similar, possibly even of the same fish at different focal planes.

We assure the reviewer that each of the images in supplemental figure 1 are distinct and represent different in situ experiments.

**Recommendations for the authors:**

**Reviewer #1 (Recommendations for the authors):**
(1) In the second sentence of the Introduction section, the acronym 'PRC' should be defined.

This has been corrected

(2) In the Discussion section, it would be useful to comment on differences between the published Xenopus cdhr1-/- OS phenotypes and the published zebrafish pcdh15b-/- OS phenotypes compared to the present zebrafish cdhr1a-/- phenotypes. In the published studies, OS in these mutants demonstrated dysmorphic and overgrown disc membranes compared to the relatively minor disc layering defects shown for cdhr1a-/- in the present study.

This discussion has been added.

(3) CDHR1 mutations in patients cause cone-rod dystrophy, but mutations in PCDH15 (Usher 1F) cause rod-cone dystrophy. In the Discussion section, the authors should comment on what might lead to these different phenotypic trajectories in humans in the context of their proposed model.

We have added to our discussion highlighting that is not possible to assess rod-cone dystrophy in the pcdh15b model as the mutation is lethal by 15dpf, which is still before most rods mature.

**Reviewer #2 (Recommendations for the authors):**
In addition to defining the 'n' for animal and cell numbers (as well as methods of analysis - automated/masked), there are a few additional recommendations for the authors.(1) Expression of USH1 genes in larval zebrafish (Figure S1) is not very convincing. SC RNAseq data exists and argues against this cell type restriction.

Based on extensive experience with WISH we are confident that our interpretation of the data are valid. Furthermore, analysis of the daniocell data base confirms that cdh23, ush1ga, ush1c (harmonin) and myo7aa all have either no expression in photoreceptors or very low levels especially compared to pcdh15b and cdhr1a.

(2) The model in Figure 1 is great. The coloring was a bit confusing. Cdhr1 and axoneme are both in green, while Pcdh15 and actin are both in red. Can each have its own color?

Changed pcdh15b color to blue

(3) Figure 2A: Please explain the multiple bands in some lanes. What do the full blots look like?

Full blots were uploaded to eLife and do not exhibit any additional bands. The multiple bands are likely due to ubiquitination or proteolytic cleavage of cdhr1a and have been documented in our previous publication (Piedade 2020).

(4) Is "data not shown" permissible? (lack of compensation of cdh1b in cdh1a mutants) (nonsense-mediated decay of the mutant transcript).

We have added a supplementary figure showcasing this data.

(5) Figure 4: Is there a TEM phenotype in discs before 15dpf? One would think there would be...?

Due to technical limitations, we have not been able to examine disc phenotypes prior to 15dpf.

(6) Figure 5: How are calyceal processes discriminated from cortical/PM-associated actin? A bonafide calyceal marker seems to be needed. Espin or Myo3, for example.

We discriminate to identify CPs as actin signal that originates at the base of the OS and travels along the OS. Pcdh15b is a bonafinde CP marker which we show overlaps with actin signal along CPs.

(7) Figures 5A-J: How is actin staining for CPs discriminating between rod and cones??? Apical - basal level imaging? This could be better clarified.

CP identification is based on co-stain for either rod or cone Oss

(8) Figure 6: Het phenotype for pcdh15b+/- (cone OS length and CP length at 5 and 10 dpf) is surprising ... worth discussing. (Figures 6E, H).

The discussion section has been updated to discuss this finding.

(9) Last, the authors state "Data not shown" throughout the manuscript. I do not believe this is allowed for the journal.

This data (cdhr1b expression in cdhr1a mutants as well as cdhr1a WISH in cdhr1a mutants) has been added as supplementary figures.

**Reviewer #3 (Recommendations for the authors):**
Major comments are addressed above and the most important is the need for a convincing demonstration of Cdhr1a localization on the outer segment and proximity to Pcdh15b. The SIM could be a powerful tool, but the images provided are impossible to assess without any basis for context. Could a membrane, Prph2, and/or actin label be added? And lower magnification views?Minor comments.(1) The mention of "short CPs" in rodents is not an accurate description. Particular rodents (e.g. mouse, rat) lack CPs altogether or have a single vestigial structure.

We have adjusted the text to reflect this point.

(2) Inconsistent spacing between numbers and units.

We have corrected these inconsistencies

(3) Missing references.

We have added missing references

(4) Indicate the mean or median for bar graphs.

The materials and methods section now specifies that all of our graphs depict a mean value

(5) Unclear how rods are distinguished from cones in the cone analysis if both are labeled with prph2 antibody.

Rods are physiological separate from cones in zebrafish retina and therefore easily identified by location as well as their distinct pattern of actin staining.

(6) Red and green should not be used together for microscopy images.(7) The diagram in Figure 1D is confusing because of the repeated use of red and green for disparate structures. Also, the location and structure of actin are misrepresented, as is the transition of disc structure during maturation in rods.

We have adjusted the color of pcdh15b to blue.